# Context-Aware Reasoner: Enhancing Contextual Reasoning in Multimodal Large Language Models

Zhe Zheng [* 1]   Wenqi Zhang [* 1]   Xiaohe Zhou [1]   Guiyang Hou [1]   Xinyu Mei [1]   Tianyang Mao [2]   Anni Piao [3]
Shengpei Jiang [3]   Yongliang Shen [1]   Yueting Zhuang [1]   Weiming Lu [1]

## Abstract

Multimodal large language models (MLLMs) have demonstrated remarkable reasoning capabilities over internalized knowledge. However, current research overlooks contextual reasoning, the ability to reason based on the relevant information present in the context. To investigate this issue, we construct the Visual Contextual Reasoning Benchmark (ContextReasonV-Bench), and our analysis reveals two predominant failure modes: *context neglect*, where models rely on pre-trained knowledge instead of contextual information, and *superficial pattern matching*, where models exploit shallow correlations rather than genuine patterns. To address these limitations, we propose a two-stage approach that progressively establishes and reinforces contextual pattern acquisition. The first stage establishes an "analyze-then-solve" reasoning paradigm through supervised fine-tuning (SFT). We then employ a context-aware reinforcement learning (RL) framework that integrates context-aware reward modeling with hierarchical advantage estimation to encourage the model to identify genuine contextual patterns. This approach yields Context-Aware Reasoner (CAR), a model that achieves 38.14% accuracy on ContextReasonV-Bench, a 22.09% improvement over the base model, and generalizes effectively to unseen tasks.

## 1. Introduction

Recent advances in large language models (LLMs) have demonstrated remarkable progress in reasoning capabilities, particularly in domains such as mathematics and code generation (DeepSeek-AI, 2025; Zhang et al., 2025b; Yu et al., 2025; Zheng et al., 2025). Among these developments, context engineering techniques have garnered widespread attention, as they enhance task performance by systematically managing and organizing contextual information from diverse input sources. For instance, in LLM-based agents for long-horizon tasks, context engineering involves designing effective memory systems that organize and manage historical interaction information for agent reasoning, including environmental states, successful trajectories, and other relevant experiences (Alizadeh et al., 2024; Xu et al., 2025a; Zhang et al., 2025a;c; Wang et al., 2025a).

However, beyond context engineering, which focuses on managing and optimizing input context to avoid context overload (Mei et al., 2025), another critical dimension is often overlooked by the community: **Can models effectively reason over the contextual information provided to them?** In practice, we frequently observe that even with a well-engineered contextual setup, models must possess the capability to comprehend cross-contextual relationships, conduct multi-hop reasoning over distributed information, and synthesize coherent conclusions from heterogeneous sources, a challenge that remains significant for current reasoning LLMs. This challenge becomes even more pronounced in multimodal scenarios, where models must not only process information distributed across text, images, and other modalities, but also reason about cross-modal relations and dependencies that may not be explicitly stated. Such contextual reasoning capability is crucial in real-world scenarios such as LLM-based agents and multi-turn dialogues, where models are expected to rapidly acquire specific skills, tool usage patterns, and users' underlying intentions from a limited number of demonstrations or unseen knowledge.

To comprehensively evaluate this capability, we construct the Visual Contextual Reasoning Benchmark (ContextReasonV-Bench), a comprehensive benchmark that systematically evaluates multimodal contextual reasoning capabilities. As illustrated in Figure 1(a), the benchmark categorizes contextual reasoning into three types based on the underlying patterns: **Rule-based reasoning** tasks, which focus on the capacity of models to identify the underly-

---

[*]Equal contribution   [1]Zhejiang University [2]Independent Researcher [3]SF Technology. Correspondence to: Weiming Lu <luwm@zju.edu.cn>.

*Proceedings of the 43rd International Conference on Machine Learning*, Seoul, South Korea. PMLR 306, 2026. Copyright 2026 by the author(s).

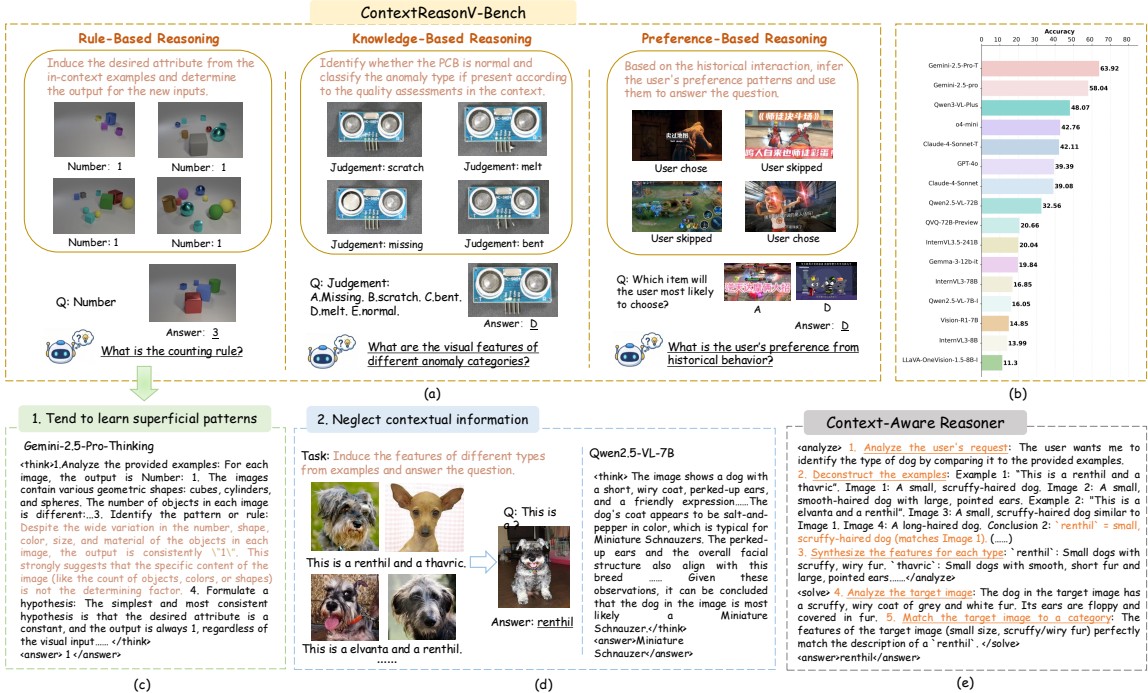

*Figure 1.* **(a)** Examples of different reasoning taxonomy in ContextReasonV-Bench. **(b)** Overall performance of representitive MLLMs on ContextReasonV-Bench. **(c)** and **(d)** demonstrate limitations of MLLMs in contextual reasoning, where (c) shows a *Context Neglect* issue on Contextual Attribute Count (CAC) task and (d) shows a *Superficial Pattern Matching* issue on Novel Concept Recognition (NCR) task. **(e)** Our CAR can effectively analyze the contextual examples and generalize the visual features corresponding to different new concepts.

ing regularities from contextual demonstrations and apply them systematically to new instances; **Knowledge-based reasoning** tasks, which evaluate whether models can accurately comprehend novel information provided in context, including new knowledge unseen during pre-training and unconventional associations between known concepts; and **Preference-based reasoning** tasks, which evaluate whether models can induce latent subjective preferences from historical interactions and apply them systematically to generate aligned responses. Each task is designed such that correct answers cannot be obtained from pre-trained knowledge alone, ensuring that evaluation reflects genuine contextual reasoning ability.

Through a comprehensive evaluation of existing MLLMs, we identify two predominant failures in contextual reasoning. *Context Neglect*: models overlook relevant contextual information and rely predominantly on internal pre-trained knowledge, accounting for 31% of errors. As shown in Figure 1 (d), the model ignores the novel conceptual definitions of objects provided in the contextual examples and instead produces responses based solely on its prior knowledge. *Superficial Pattern Matching*: models fail to accurately interpret contextual information, or rely on shallow surface-level correlations without capturing the underlying patterns. As illustrated in Figure 1 (c), the model is confused by multi-

ple identical counts in the examples, failing to effectively identify the genuine associative patterns between objects and counts. These failure modes persist even in models explicitly trained to enhance reasoning capabilities. Rule-based RL models such as Vision-R1-7B (Huang et al., 2025) and MM-Eureka-7B (Meng et al., 2025) achieve substantial gains on mathematical and scientific benchmarks, yet show limited improvement over base models on contextual reasoning tasks. Therefore, it is necessary to design targeted methods to enhance models' contextual reasoning abilities.

We develop Context-Aware Reasoner (CAR) with a carefully designed two-stage training strategy. To address the issue of context neglect, we first construct high-quality reasoning data to perform supervised fine-tuning (SFT), enabling the model to learn the reasoning paradigm of analyzing and extracting relevant contextual information. For the problem of superficial pattern matching, we employ a context-aware enhanced RL approach to further elicit the model's deep reasoning capabilities, where context-aware reward is designed to encourage in-depth exploration of the underlying authentic patterns in the context. We employ hierarchical advantage estimation, that constructs semantically equivalent context variants and estimates advantages both within and across variants, to enhance the robustness of contextual reasoning and mitigate interference from semantically

irrelevant contextual factors.

Extensive experiments demonstrate that CAR yields significant performance improvements. CAR achieves an overall accuracy of 38.14% on ContextReasonV-Bench, representing a 22.09% improvement over the base model, with gains of 11.83%, 28.57%, and 20.34% on rule-based, knowledge-based, and preference-based reasoning tasks respectively. CAR also generalizes well to tasks not seen during training, confirming that our approach enhances genuine contextual reasoning rather than overfitting to specific task formats. Our main contributions can be summarized as follows:

- We reveal fundamental limitations of existing MLLMs in context reasoning, demonstrating that they tend to overlook contextual information and engage in superficial pattern matching.

- We construct ContextReasonV-Bench, which is designed to systematically evaluate multimodal contextual reasoning capabilities under three representative latent interaction patterns.

- We propose a two-stage training approach that integrates context-aware reward modeling with hierarchical advantage estimation to boost the model's contextual reasoning capability. Comprehensive experiments demonstrate the promising contextual reasoning and robust generalization of CAR.

## 2. ContextReasonV-Bench

### 2.1. Task Formulation

Given a set of context examples $\mathcal{C} = \{(x_i, y_i)\}_{i=1}^{K}$ and a query $\mathcal{Q}$, where $x_i$ denotes visual or interleaved image-text input and $y_i$ denotes the corresponding output, which can be an empty set in some tasks, contextual reasoning requires the model to produce a response $\mathcal{Y}$ by extracting and applying patterns from $\mathcal{C}$ to $\mathcal{Q}$:

$$\mathcal{Y} = \pi_\theta(\mathcal{C}, \mathcal{Q}) \tag{1}$$

where $\pi_\theta$ is the policy model. Crucially, each task is designed such that $\mathcal{Y}$ cannot be correctly inferred from $\mathcal{Q}$ alone using pre-trained knowledge; the model must leverage $\mathcal{C}$ to succeed. Based on the nature of patterns that must be extracted from context, we categorize contextual reasoning into three types: rule-based, knowledge-based, and preference-based reasoning.

**Rule-based Reasoning.** This category requires models to induce transformation rules from input-output demonstrations and apply them to new instances. The context provides examples of a latent rule $\mathcal{R}$ governing the mapping from inputs to outputs. The reasoning process involves rule induction followed by rule application:

$$\mathcal{Y} = \mathcal{H}(\mathcal{I}_\mathcal{R}(\mathcal{C}), \mathcal{Q}) \tag{2}$$

where $\mathcal{I}_\mathcal{R} : \mathcal{C} \rightarrow \mathcal{R}$ denotes the rule induction function that extracts the underlying rule from context examples, and $\mathcal{H}$ applies the induced rule to the query. For example, in the Contextual Attribute Counting task (Figure 1(a)), models observe image-count pairs and must infer which visual attribute (color, shape, or size) determines the count, then apply this rule to count objects in new images.

**Knowledge-based Reasoning.** This category evaluates whether models can acquire and apply novel information from context, including new concepts unseen during pre-training and unconventional associations between known entities. Given context examples containing relevant knowledge, the model must retrieve and apply the most pertinent information:

$$\mathcal{Y} = \mathcal{H}(k^*, \mathcal{Q}), \quad k^* = \arg\max_{k \in \mathcal{K}} \mathrm{Rel}(k, \mathcal{Q}) \tag{3}$$

where $\mathcal{K} = \mathcal{I}_\mathcal{K}(\mathcal{C})$ is the knowledge set induced from context, $k^*$ is the knowledge most relevant to the query, and $\mathrm{Rel}(\cdot)$ measures relevance. We further distinguish two knowledge types: *procedural knowledge* specifying how to perform tasks (e.g., multi-step assembly instructions), and *declarative knowledge* specifying factual or conceptual information (e.g., visual characteristics of anomaly categories). For instance, in the Contextual Anomaly Identification task (Figure 1(b)), models must induce the visual features of each anomaly type from labeled examples and apply this knowledge to identify new instances.

**Preference-based Reasoning.** This category requires models to infer latent subjective preferences from historical interactions and apply them to generate aligned responses or predictions. Given a user's interaction history as context, the model must extract a preference function and employ it to evaluate new candidates:

$$\mathcal{Y} = \mathcal{H}(\Gamma_p, \mathcal{Q}) \tag{4}$$

where $\Gamma_p = \mathcal{I}_\mathcal{P}(\mathcal{C})$ denotes the preference function induced from context. This requires discriminating preference-critical features through contrastive analysis of chosen versus rejected items. For example, in the Preference-aware Item Recommendation task (Figure 1(c)), models observe a user's interaction history and must infer which item the user will choose next.

### 2.2. Benchmark Construction

We design 16 tasks spanning the three reasoning categories, covering diverse real-world scenarios including visual pattern recognition, procedural instruction following, anomaly detection, sign language understanding, and personalized recommendation. Each task requires multi-hop inference and cross-example pattern integration that cannot be solved through simple template matching.

*Table 1.* Performance of representitive MLLMs on ContextReasonV-Bench

| Model | Rule-Based Reasoning | | | | Knowledge-Based Reasoning | | | | | | | | | | | Preference-Based Reasoning | | | | Overall |
| | CTT | VLT | CAC | Avg. | CVU | LIU | CIOD | NCR | CSLR | CAI | SRI | MC | STL | AD | Avg. | CPI | PIS | PIR | Avg. | |
|---|---|---|---|---|---|---|---|---|---|---|---|---|---|---|---|---|---|---|---|---|
| Random | - | 12.50 | 18.60 | - | 16.00 | 20.00 | 27.67 | 0.00 | - | 15.48 | - | - | 25.00 | - | - | 20.00 | 33.33 | 25.00 | - | - |
| w/o context | 1.00 | 10.94 | 3.87 | - | 8.00 | 13.33 | 20.67 | 0.00 | 5.64 | 8.47 | 6.00 | 0.00 | 15.00 | 2.00 | - | 19.00 | 13.00 | 16.00 | - | - |
| Human | 100.00 | 80.16 | 88.37 | 88.73 | 100.00 | 100.00 | 93.33 | 94.67 | 90.75 | 88.98 | 100.00 | 85.00 | 100.00 | 100.00 | 94.42 | 87.00 | 90.00 | 85.00 | 87.33 | 91.23 |
| *Close-Source MLLMs* | | | | | | | | | | | | | | | | | | | | |
| Gemini-2.5-Pro-Thinking(Team, 2025a) | 69.00 | 67.46 | 65.12 | 67.04 | 84.00 | 80.00 | 60.67 | 76.00 | 45.37 | 55.09 | 22.00 | 75.00 | 90.00 | 73.13 | 64.33 | 62.00 | 51.00 | 65.00 | 59.33 | 63.92 |
| Claude-4-Sonnet-Thinking(Anthropic, 2025) | 56.00 | 43.65 | 51.94 | 50.14 | 42.00 | 60.00 | 37.33 | 30.67 | 20.68 | 31.36 | 16.00 | 55.00 | 45.00 | 52.83 | 35.39 | 60.00 | 51.00 | 30.00 | 47.00 | 42.11 |
| o4-mini(OpenAI, 2025) | 47.00 | 61.11 | 44.19 | 50.99 | 60.00 | 73.33 | 29.33 | 27.33 | 55.19 | 48.30 | 18.00 | 55.00 | 75.00 | 40.00 | 38.73 | 61.00 | 32.00 | 32.00 | 41.67 | 42.76 |
| Gemini-2.5-pro(Team, 2025a) | 75.00 | 60.32 | 70.54 | 68.17 | 84.00 | 80.00 | 60.00 | 42.67 | 41.70 | 49.15 | 10.00 | 65.00 | 75.00 | 34.01 | 50.44 | 67.00 | 59.00 | 61.00 | 62.33 | 58.04 |
| Claude-4-Sonnet(Anthropic, 2025) | 59.00 | 43.65 | 47.29 | 49.30 | 42.00 | 46.67 | 27.33 | 28.67 | 25.93 | 24.58 | 12.00 | 50.00 | 35.00 | 44.04 | 29.74 | 64.00 | 43.00 | 34.00 | 47.00 | 39.08 |
| GPT-4o(OpenAI, 2024) | 34.00 | 27.78 | 34.88 | 32.11 | 64.00 | 80.00 | 50.67 | 52.67 | 17.25 | 43.22 | 8.00 | 60.00 | 30.00 | 51.55 | 46.85 | 32.00 | 33.00 | 31.00 | 32.00 | 39.39 |
| Qwen3-VL-Plus(Team, 2025c) | 65.00 | 67.46 | 75.19 | 69.58 | 52.00 | 86.67 | 36.00 | 28.00 | 34.65 | 28.81 | 10.00 | 55.00 | 90.00 | 6.00 | 33.12 | 60.00 | 47.00 | 57.00 | 54.67 | 48.07 |
| *Open-Source MLLMs* | | | | | | | | | | | | | | | | | | | | |
| QVQ-72B-Preview(Qwen Team, 2024) | 12.00 | 30.16 | 16.28 | 20.00 | 26.00 | 26.67 | 22.67 | 2.67 | 5.92 | 9.32 | 8.00 | 45.00 | 30.00 | 29.94 | 15.73 | 36.00 | 25.00 | 35.00 | 32.00 | 20.66 |
| Qwen2.5-VL-72B(Bai et al., 2025a) | 45.00 | 27.78 | 16.28 | 28.45 | 32.00 | 46.67 | 28.00 | 48.67 | 21.68 | 25.42 | 14.00 | 50.00 | 30.00 | 68.62 | 35.71 | 31.00 | 34.00 | 27.00 | 30.67 | 32.56 |
| InternVL3.5-241B(Wang et al., 2025b) | 6.00 | 38.89 | 13.95 | 20.56 | 18.00 | 20.00 | 22.00 | 8.00 | 14.13 | 26.27 | 6.00 | 25.00 | 50.00 | 37.57 | 19.92 | 6.00 | 15.00 | 38.00 | 19.67 | 20.04 |
| InternVL3-78B(Zhu et al., 2025) | 10.00 | 34.92 | 14.73 | 20.56 | 15.00 | 13.33 | 19.33 | 3.00 | 6.83 | 11.87 | 8.00 | 25.00 | 35.00 | 58.60 | 16.12 | 7.00 | 13.00 | 22.00 | 14.00 | 16.85 |
| Gemma-3-12b-it(Team, 2025b) | 31.00 | 32.54 | 5.43 | 22.25 | 22.00 | 26.67 | 12.00 | 12.00 | 10.38 | 24.58 | 4.00 | 25.00 | 35.00 | 5.00 | 15.33 | 25.00 | 27.00 | 28.00 | 26.67 | 19.84 |
| LLaVA-OneVision-1.5-8B-Instruct(An et al., 2025) | 6.00 | 23.81 | 2.33 | 10.99 | 12.00 | 6.67 | 5.33 | 1.33 | 3.33 | 11.02 | 10.00 | 30.00 | 30.00 | 2.00 | 7.57 | 23.00 | 24.00 | 12.00 | 19.67 | 11.30 |
| InternVL3-8B(Zhu et al., 2025) | 5.00 | 24.60 | 11.63 | 14.37 | 14.00 | 20.00 | 4.00 | 2.67 | 7.92 | 5.93 | 2.00 | 40.00 | 25.00 | 2.00 | 6.78 | 30.00 | 30.00 | 27.00 | 29.00 | 13.99 |
| Qwen2.5-VL-7B-Instruct(Bai et al., 2025a) | 12.00 | 25.40 | 5.43 | 14.37 | 16.00 | 13.33 | 23.33 | 10.00 | 6.92 | 6.78 | 4.00 | 20.00 | 35.00 | 15.94 | 14.05 | 23.00 | 25.00 | 19.00 | 22.33 | 16.05 |
| *Rule-based RL MLLMs* | | | | | | | | | | | | | | | | | | | | |
| MM-Eukera-7B(Meng et al., 2025) | 7.00 | 26.98 | 6.20 | 13.80 | 20.00 | 20.00 | 25.33 | 12.00 | 12.96 | 8.47 | 6.00 | 20.00 | 35.00 | 6.00 | 15.33 | 21.00 | 29.00 | 15.00 | 21.67 | 16.38 |
| Vision-R1-7B(Huang et al., 2025) | 8.00 | 28.57 | 9.30 | 15.77 | 22.00 | 33.33 | 17.33 | 1.33 | 9.00 | 5.09 | 4.00 | 35.00 | 30.00 | 2.00 | 10.86 | 21.00 | 29.00 | 17.00 | 22.33 | 14.85 |

**Design Principles.** We adhere to three principles in benchmark construction. First, *context dependency*: each instance must require contextual information to solve, ensuring that pre-trained knowledge alone is insufficient. Second, *pattern authenticity*: context examples must contain genuine, unambiguous patterns that humans can reliably identify. Third, *interference resistance*: we deliberately include distractors such as irrelevant visual features or coincidental correlations to test whether models capture true underlying patterns.

**Data Filtering.** To ensure strict context dependency, we filter samples that can be correctly answered without context. Specifically, we remove context from each sample and prompt GPT-4o to answer queries directly. To account for random guessing in multiple-choice questions, we conduct three independent evaluations and exclude samples answered correctly in all trials. This filtering removes instances where pre-trained knowledge provides shortcuts to the correct answer.

**Quality Control.** We conduct human evaluation on the benchmark. Three expert annotators assess samples along three dimensions: pattern uniqueness (whether a single clear pattern exists), pattern clarity (whether the pattern is unambiguously identifiable), and information sufficiency (whether context provides adequate information for correct inference). Samples failing any criterion are revised or removed.

Table 6 summarizes the benchmark statistics. The 16 tasks distribute across three categories: 3 rule-based tasks (Contextual Text Transformation, Visual Logical Transformation, Contextual Attribute Counting), 10 knowledge-based tasks covering both procedural knowledge (Cross View Understanding, Lego Instruction Understanding) and declarative knowledge (Novel Concept Recognition, Contextual Sign Language Recognition, Contextual Anomaly Identifica-

tion, etc.), and 3 preference-based tasks (Caption Preference Inference, Preference-based Image Selection, Preference-aware Item Recommendation). Detailed task descriptions are provided in Appendix A.

## 2.3. Evaluation and Analysis

We conduct comprehensive experiments on state-of-the-art MLLMs to systematically evaluate their contextual reasoning capabilities, including proprietary MLLMs, open-source MLLMs and rule based RL models. The detailed results are shown in Table 1.

### 2.3.1. OVERALL PERFORMANCE

**Proprietary Model Performance.** Among proprietary models, Gemini-2.5-Pro-Thinking achieves the highest overall score (63.92%).Other multimodal reasoning models, Claude-4-Sonnet-Thinking and o4-mini achieve accuracies of 42.11% and 42.76% respectively. Gemini-2.5-Pro, GPT-4o, Claude-4-Sonnet and Qwen3-VL-Plus obtained scores of 58.04%, 39.39%, 39.08% and 48.07% respectively. Among the three categories of contextual reasoning, proprietary models generally demonstrate superior performance in rule-based reasoning, compared to knowledge-based and preference-based reasoning. This advantage can be attributed to the fact that rule-based reasoning possesses a more explicit logical structure and deterministic nature, which is more closely aligned with the logical reasoning capabilities learned during model training.

**Open-source Model Performance.** For open-sourced models, Qwen2.5-VL-72B-Instruct achieves an average score of 32.56%, while all other models fall significantly short with scores below 21%, indicating that existing open-sourse MLLMs exhibit significant limitations in contextual reasoning capabilities. At the subtype level, several tasks

prove challenging for nearly all models. For instance, Sign Language Recognition and Contextual Rotation tasks yield uniformly low accuracy across the board, with the best performances reaching only 21.68% and 14.00%, respectively. These results suggest that existing open-source models struggle to effectively integrate and generalize contextual information, especially when tasks demand the capture of fine-grained relationships and exploitation of contextual dependencies beyond surface-level pattern matching.

**Rule based RL Model Performance.** The rule based RL models Vision-R1-7B and MM-Eukera-7B achieve performance scores of 14.85% and 16.38%, respectively, demonstrating negligible improvement compared to their base model Qwen2.5-VL-7B (16.05%). Specifically, Vision-R1-7B shows a decline of 3.19% and 1.2% on knowledge-based and preference-based reasoning tasks compared to Qwen2.5-VL-7B. While RL training on large-scale mathematical reasoning data effectively enhances long-chain reasoning in formal domains, it fails to improve contextual reasoning abilities. This deficiency can be attributed to the substantial domain gap between the mathematical reasoning tasks and diverse contextual reasoning scenarios, where mathematical reasoning primarily involves deterministic rule application with pre-trained knowledge, whereas contextual reasoning demands flexible integration of heterogeneous information. Therefore, developing effective training paradigms to enhance contextual reasoning capabilities of MLLMs holds significant importance.

### 2.3.2. Error Analysis

To identify the primary challenges of contextual reasoning, we perform a comprehensive error analysis on representative MLLMs. We find that models tend to overlook relevant contextual information and rely predominantly on their internal pre-trained knowledge. For example, in Figure 1(d), the model disregards the novel conceptual definitions of objects provided in the contextual examples and instead responds directly based on its internal knowledge. This behavior is fundamentally misaligned with the task objective, revealing the model's inability to effectively acquire and integrate information from context. Besides, models frequently fail to accurately comprehend contextual information or engage in superficial pattern matching without capturing implicit relational patterns. As illustrated in Figure 1(c), the model is confused by multiple identical counts appearing in the contextual examples, failing to effectively identify the genuine associative patterns between objects and counts. We refer to the two aforementioned failure phenomena as Context Neglect issue and Superficial Pattern Matching issue.

To further quantify error patterns in the reasoning process, we randomly select 200 erroneous samples from Qwen2.5-VL-7B and analyze their reasoning processes. In rule-based

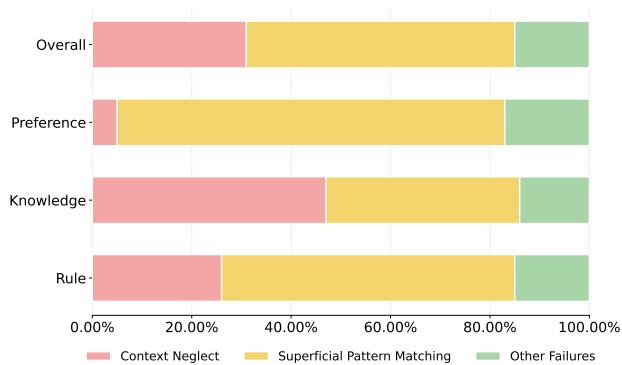

*Figure 2.* Error Distribution of Different Reasoning Tasks.

reasoning tasks, approximately 59% of the errors can be attributed to Superficial Pattern Matching, while 26% result from Context Neglect. In knowledge-based reasoning tasks, Context Neglect becomes more prominent, accounting for 47% of the errors, whereas Superficial Pattern Matching constitutes 39%. In preference-based reasoning tasks, Superficial Pattern Matching dominates error distribution at 78%, as preferences are typically multi-dimensional and fine-grained, discriminating preference-critical features from context remains a challenge. The remaining errors arise from issues such as misapplying induced patterns and generating answers inconsistent with the reasoning.

## 3. Training Recipes for CAR

We employ a carefully designed two-stage training strategy to enhance contextual reasoning capabilities. To address the issue of context neglect, we first construct high-quality reasoning data for SFT, enabling the model to learn the reasoning paradigm of analyzing and extracting relevant contextual information. We then leverage a context-aware RL approach (illustrated in Figure 3) to further elicit the model's deep reasoning capabilities. Specifically, we utilize Group Relative Policy Optimization (GRPO) for stable policy optimization and incorporate context-aware rewards to encourage in-depth exploration of the underlying authentic patterns in the context. Furthermore, we adopt hierarchical advantage estimation to guide the model to focus on contextual semantic information, mitigating interference from irrelevant factors, and thus achieving more robust reasoning.

### 3.1. Learning Context-Aware Reasoning Patterns from High-Quality Supervision

**High-Quality Data Construction.** To enable the model to learn the reasoning paradigm of analyzing contextual information before problem-solving, we construct a curated dataset of high-quality reasoning chains for supervised fine-tuning. Specifically, we first select representative

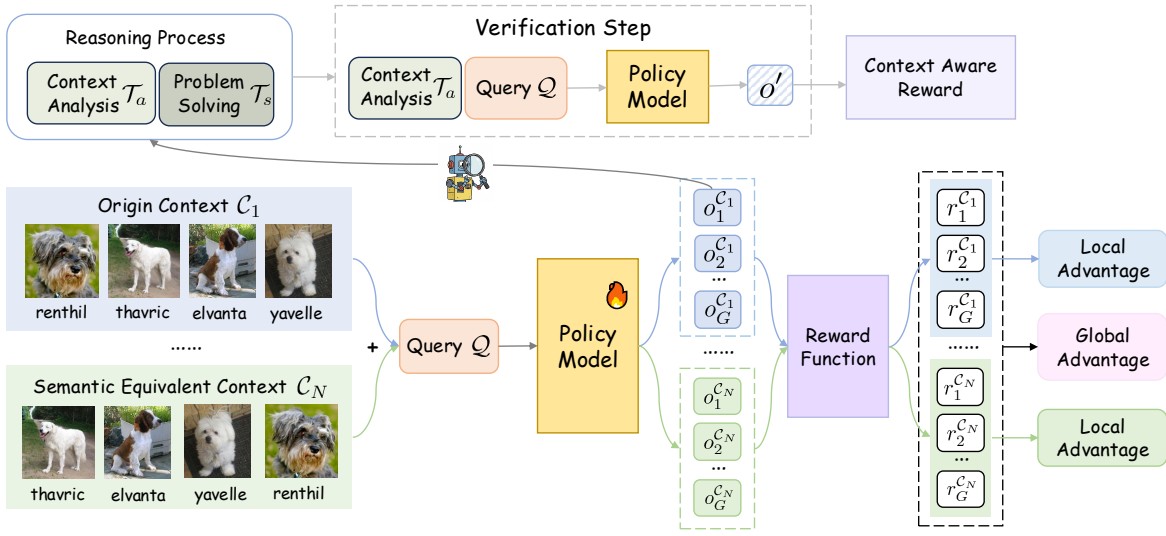

*Figure 3.* **Context-Aware RL Framework.** We design a context-aware reward that explicitly supervises the model's intermediate reasoning process by validating the effectiveness of captured contextual patterns. We employ hierarchical advantage estimation to enhance the robustness and generalization of contextual reasoning.

contextual reasoning tasks from established benchmarks and construct the corresponding training data. For each candidate task, we randomly sample one instance from training dataset and manually construct a reasoning chain following the "context analysis then problem-solving" paradigm. The reasoning process is formalized as: `<analyze>...</analyze><solve>...</solve> <answer>...</answer>`. These manually crafted examples serve as few-shot demonstrations to prompt Gemini-2.5-Pro-Thinking to generate reasoning chains following the same paradigm for the remaining training instances. Finally, we filter the generated data by retaining only those with correct answers. The final training dataset comprises 388 high-quality samples, with detailed data distribution shown in Figure 7.

### 3.2. Context-Aware RL with Hierarchical Advantage Estimation

#### 3.2.1. REWARD MODELING

**Context Aware Reward.** To address the challenge that models often fail to capture relevant contextual information or merely exploit superficial contextual patterns, we design a context-aware reward $r_c$ that explicitly supervises the model's intermediate reasoning process by validating the effectiveness of captured contextual patterns. Building upon the reasoning paradigm learned during the SFT stage, the model's reasoning process can be decomposed into two distinct phases: context analysis $\mathcal{T}_a$ and problem-solving $\mathcal{T}_s$, with the final generated answer denoted as $\mathcal{A}$. To val-

idate whether the model has effectively extracted relevant contextual information, we replace the original multi-modal context $\mathcal{C}$ with the model's generated context analysis $\mathcal{T}_a$, and prompt the model to solve the problem based solely on this extracted information. The reasoning and verification steps are defined as follows:

$$\begin{aligned} \text{Reasoning Step:} \quad & <\mathcal{T}_a, \mathcal{T}_s, \mathcal{A}> = \pi_\theta(\mathcal{C}, \mathcal{Q}) \\ \text{Verification Step:} \quad & <\mathcal{T}_v, \mathcal{A}_v> = \pi_\theta(\mathcal{T}_a, \mathcal{Q}) \end{aligned} \quad (5)$$

where $\pi_\theta$ represents MLLM, $Q$ is the query, $\mathcal{T}_v$ and $\mathcal{A}_v$ represent the reasoning chains and the final answer of the verification step.

The context-aware reward $r_c$ is granted if and only if the verification step yields the correct answer, which serves as a strong indicator that the model has successfully extracted the essential contextual patterns necessary for problem-solving, demonstrating that the context analysis contains sufficient information to reach the correct answer.

$$r_c = \begin{cases} 1, & \text{if } \mathcal{A}_v \text{ is true,} \\ 0, & \text{otherwise.} \end{cases} \quad (6)$$

**Total Reward.** The final reward function consists of three components: format reward $r_{\text{format}}$, accuracy reward $r_{\text{acc}}$ and context aware reward $r_c$, which is formulated as follows:

$$r = \alpha \cdot r_{\text{format}} + r_{\text{acc}} + r_c \cdot r_{\text{acc}} \quad (7)$$

We set the format reward coefficient as $\alpha = 0.5$ in our experiment. To avoid rewarding coincidentally correct but un-

informative contextual analyses, we introduce a correctness-gating mechanism that grants the context-aware reward only when the model answers correctly in both the reasoning and verification steps, preventing superficial contextual patterns from being reinforced.

### 3.2.2. HIERARCHICAL ADVANTAGE ESTIMATION

To enhance the robustness and generalization of contextual reasoning while mitigating interference from semantically irrelevant contextual factors, such as the order of in-context examples or superficial formatting variations (Rieff et al., 2025), we propose hierarchical advantage estimation. This approach augments the contextual semantic space by constructing semantically equivalent context variants, and estimates reasoning trajectory advantages across and within context variant.

**Context Semantic Augmentation.** To construct semantically equivalent context variants while preserving the essential information content, we employ augmentation strategies tailored to the structure of the context. For multi-example contexts, we generate variants by randomly permuting the order of demonstration examples. Each permutation maintains the same set of examples but presents them in different sequences. For single-example contexts, we perform visual augmentation of context images. Specifically, we permute the spatial arrangement of step panels for multi-step instructional images, while preserving the correspondence between each step's visual content, textual description, and identifier to ensure semantic consistency. For other images, we apply rotation transformations to create variants. The expanded context space is denoted as $\mathbf{C} = \{\mathcal{C}_1, \mathcal{C}_2, \cdots, \mathcal{C}_N\}$.

**Advantage Estimation.** We estimate the advantage of each reasoning trajectory by jointly considering relative advantages at both the local and global levels. Specifically, for each context variant $\mathcal{C}_j$ with its query $\mathcal{Q}$, the policy model $\pi_\theta$ generates $G$ candidate reasoning responses, denoted as $\{o_1^{\mathcal{C}_j}, o_2^{\mathcal{C}_j}, \cdots, o_G^{\mathcal{C}_j}\}$. We adopt rule-based reward functions to compute the reward for each reasoning trajectory. Let $\mathbf{R} = \{R^{\mathcal{C}_1}, \cdots, R^{\mathcal{C}_N}\}$ denote the set of rewards across all context variants, where $R^{\mathcal{C}_j} = \{r_1^{\mathcal{C}_j}, \ldots, r_G^{\mathcal{C}_j}\}$ represents the rewards corresponding to the $G$ trajectories generated under context $\mathcal{C}_j$. To capture both intra-context and inter-context comparisons, we employ a hierarchical advantage estimation strategy that combines the relative advantage within the same context (local level) and the relative advantage across all semantic context variants (global level). The final advantage for the $i$-th trajectory under context $\mathcal{C}_j$ is computed as follow:

$$A_i^j = \frac{r_i^{\mathcal{C}_j} - \mathrm{mean}(R^{\mathcal{C}_j})}{\mathrm{std}(R^{\mathcal{C}_j})} + \frac{r_i^{\mathcal{C}_j} - \mathrm{mean}(\{R^{\mathcal{C}_1}, \cdots, R^{\mathcal{C}_N}\})}{\mathrm{std}(\{R^{\mathcal{C}_1}, \cdots, R^{\mathcal{C}_N}\})}$$

(8)

Hierarchical advantage estimation explicitly encourages the

model to learn reasoning patterns that are invariant to irrelevant surface variations. Consequently, this method improves the model's ability to focus on semantically essential contextual information while remaining robust to spurious variations in context.

The loss function for the RL phase is defined as follows:

$$J(\theta) = \mathbb{E}_{\{o_i\}_{i=1}^G} \left[ \frac{1}{G} \sum_{i=1}^G \frac{1}{N} \sum_{j=1}^N \min \left( \frac{\pi_\theta(o_i^{\mathcal{C}_j}|\mathcal{C}_j, \mathcal{Q})}{\pi_{\theta_{\mathrm{old}}}(o_i^{\mathcal{C}_j}|\mathcal{C}_j, \mathcal{Q})} A_i, \right. \right.$$
$$\left. \left. \mathrm{clip} \left( \frac{\pi_\theta(o_i^{\mathcal{C}_j}|\mathcal{C}_j, \mathcal{Q})}{\pi_{\theta_{\mathrm{old}}}(o_i^{\mathcal{C}_j}|\mathcal{C}_j, \mathcal{Q})}, 1-\epsilon, 1+\epsilon \right) A_i^j \right) \right]$$

(9)

where $\pi_{\theta_{\mathrm{old}}}$ represents the policy model and $\epsilon$ is a clipping hyper-parameter.

## 4. Experiments

### 4.1. Implementation Details

In this work, we employ Qwen2.5-VL-7B-Instruct(Bai et al., 2025b) as base model. The distribution of the training data are presented in Table 7. We train the model for 3 epochs via SFT, followed by 250 steps for RL. We use a context variant $N$ as 2 and a sample number $G$ as 8 per query in RL phase. More training details can be seen in Appendix C.

### 4.2. Main Results

Table 2 presents comprehensive evaluation results on ContextReasonV-Bench. Compare to the base model Qwen2.5-VL-7B, CAR-7B achieves substantial performance improvements on contextual reasoning tasks (+22.09% overall), with gains of 11.83%, 28.57%, and 20.34% on rule-based, knowledge-based, and preference-based reasoning tasks respectively. The performance gain is particularly pronounced on challenging tasks like Contextual Text Transformation (+29%), Novel Concept Recognition (+63.33%), and Preference-aware Item Recommendation (+33%). The CAR-SFT trained in the first stage demonstrates notable improvements on in-domain tasks. For example, the CAR-SFT achieves improvements of +24% and +15.34% on Cross View Understanding and Contextual Image Operation Deduction tasks, validating that high-quality reasoning training data enables the model to effectively learn context-aware reasoning patterns and alleviates the context neglect problem. However, its performance degrades on out-of-distribution tasks, especially those where the contextual format differs significantly from the training data, with decreases of 2.39% and 15.00% on Visual Logical Transformation and Symbolic Transformation Learning tasks.

**Generalization Analysis.** The results demonstrate that CAR exhibits strong generalization capabilities on tasks

*Table 2.* **Main Results.** We examine our CAR-7B across various contextual reasoning tasks. **Bold** numbers indcate the best performance. Tasks marked with purple are not included in the training data, and tasks marked with blue include sub-tasks unseen in the training. Improvement denotes the performance gain of CAR-7B over the base model.

| Model | Data Size | Rule-Based Reasoning | | | | Knowledge-Based Reasoning | | | | | | | | | | | Preference-Based Reasoning | | | | Overall |
|---|---|---|---|---|---|---|---|---|---|---|---|---|---|---|---|---|---|---|---|---|---|
| | | CTT | VLT | CAC | Avg. | CVU | LIU | CIOD | NCR | CSLR | CAI | SRI | MC | STL | AD | Avg. | CPI | PIS | PIR | Avg. | |
| MM-Eukera-7B | 15k | 7.00 | 26.98 | 6.20 | 13.80 | 20.00 | 20.00 | 25.33 | 12.00 | 12.96 | 8.47 | 6.00 | 20.00 | 35.00 | 6.00 | 15.33 | 21.00 | 29.00 | 15.00 | 21.67 | 16.38 |
| Vision-R1-7B | 210k | 8.00 | **28.57** | 9.30 | 15.77 | 22.00 | 33.33 | 17.33 | 1.33 | 9.00 | 5.09 | 4.00 | **35.00** | 30.00 | 2.00 | 10.86 | 21.00 | 29.00 | 17.00 | 22.33 | 14.85 |
| Qwen2.5-VL-7B-Instruct | - | 12.00 | 25.40 | 5.43 | 14.37 | 16.00 | 13.33 | 23.33 | 10.00 | 6.92 | 6.78 | 4.00 | 20.00 | 35.00 | 15.94 | 14.05 | 23.00 | 25.00 | 19.00 | 22.33 | 16.05 |
| CAR-SFT | 0.4k | 28.00 | 23.01 | 9.30 | 19.43 | 40.00 | 13.33 | 38.67 | 52.67 | 6.79 | 9.00 | 6.00 | 20.00 | 15.00 | 23.96 | 30.38 | 25.00 | 33.00 | 37.00 | 31.67 | 27.68 |
| **CAR-7B** | 1.7k | **41.00** | 27.78 | **13.18** | **26.20** | **56.00** | **26.67** | **46.67** | **73.33** | **13.18** | **15.25** | **20.00** | **35.00** | **50.00** | **28.76** | **42.62** | **34.00** | **42.00** | **52.00** | **42.67** | **38.14** |
| Improvement | - | 29.00 | 2.38 | 7.75 | 11.83 | 40.00 | 13.34 | 23.34 | 63.33 | 6.26 | 8.47 | 16.00 | 15.00 | 15.00 | 12.82 | 28.57 | 11.00 | 17.00 | 33.00 | 20.34 | 22.09 |

*Table 3.* Results on General Vision Tasks

| Model | MMMU-Pro | MMMU | VisNum Bench | BLINK | MMStar | MathVerse | MATH-Vision | Hallusion Bench | Avg. |
|---|---|---|---|---|---|---|---|---|---|
| Qwen2.5-VL-7B | 41.97 | 53.78 | 29.52 | 45.10 | 62.07 | 43.55 | 24.07 | 64.57 | 45.58 |
| CAR-7B | 49.69 | 55.56 | 36.07 | 51.16 | 63.67 | 44.82 | 27.46 | 68.40 | 49.60 |

not included in training. For instance, the model achieves performance improvements of 15% and 12.82% on Symbolic Transformation Learning (STL) task and Alphabet Decoding (AD) task. Notably, on the challenging Preference-aware Item Recommendation (PIR) task, the model achieves a substantial improvement of 33%. Despite being trained only on the relatively straightforward Preference-based Image Selection (PIS) task, the model successfully transfers its preference-based reasoning capabilities to more complex and diverse scenarios. These findings validate that our training approach effectively enhances the model's ability to perform robust and generalizable contextual reasoning in complex scenarios.

### 4.3. Evaluations on General Tasks

We evaluate the model on general vision tasks, including general visual understanding (i.e., MMMU-Pro (Yue et al., 2025), MMMU (Yue et al., 2024), VisNumBench (Weng et al., 2025), MMStar(Chen et al., 2024), BLINK(Fu et al., 2024)), multimodal math reasoning (i.e., MathVista(Lu et al., 2024), MathVerse(Zhang et al., 2024)) and hallucination diagnosis (Guan et al., 2024). As shown in Table 3, our model achieves an average improvement of 4.74% on general visual understanding tasks and 2.33% on visual math reasoning tasks. Notably, our model demonstrates substantial gains on MMMU-Pro (+7.72%), BLINK (+6.06%), and HallusionBench (+3.83%), suggesting that our approach effectively enhances the model's ability to leverage visual information within context and mitigates hallucinations in cross-modal reasoning. More analysis can be seen in D.2.

### 4.4. Ablation Study

As shown in Table 4, we conduct ablation studies to examine the individual contribution of each design in the training method, including context-aware reward and hierarchical advantage estimation.

**Effective of context aware reward.** Compared to the vanilla GRPO baseline, the model trained with context-aware reward alone (i.e., w/o HAE & CSA) achieves a 1.39% improvement. This validates that incorporating context-aware rewards in the RL stage enables the model to capture more accurate contextual information and patterns.

**Effective of hierarchical advantage.** The introduction of hierarchical advantage estimation yields a substantial performance gain of 3.55%. By combining the relative advantages of reasoning trajectories both within the same context and across semantically equivalent context variants, this approach encourages the model to focus on genuine semantic patterns while reducing the influence of irrelevant surface-level variations, thereby enhancing robustness and generalization in contextual reasoning. To disentangle the contribution of context semantic augmentation from that of advantage estimation, we train a model variant using augmented data without hierarchical advantage estimation (i.e., w/o HAE). Results show that the increased data scale from context augmentation alone contributes a 2.15% performance improvement.

*Table 4.* **Ablation Experimental Results.** HAE and CSA represent hierarchical advantage estimation and context aware reward.

| Method | Rule | Knowledge | Preference | Overall |
|---|---|---|---|---|
| **CAR-7B** | **26.20** | **42.62** | **42.67** | **38.14** |
| w/o HAE | 25.07 | 40.99 | 41.33 | 36.74 |
| w/o HAE & CSA | 21.97 | 39.15 | 39.67 | 34.59 |
| Vanilla GRPO | 20.85 | 37.77 | 38.00 | 33.20 |

## 5. Conclusions

In this work, we explore the contextual reasoning capabilities of MLLMs. We introduce ContextReasonV-Bench to comprehensively evaluate multimodal contextual reasoning

capabilities and reveal that existing MLLMs often neglect critical contextual information and fail to capture genuine underlying patterns. We design a two-stage training approach to enhance contextual reasoning capabilities. We first construct high-quality training data for SFT, enabling the model to learn an "analyze-then-solve" paradigm that addresses the context neglect problem. In the second stage, we employ context-aware rewards and hierarchical advantage estimation to encourage the model to focus on capturing genuine contextual patterns. Our CAR model achieves great performance improvement on contextual reasoning tasks and demonstrates robust generalization capabilities.

## Acknowledgments

This work is supported by the National Natural Science Foundation of China (No. 62376245), the Key Research and Development Program of Zhejiang Province, China (No. 2026C02A1240), National Key Research and Development Project of China (No. 2018AAA0101900), and MOE Engineering Research Center of Digital Library.

## Impact Statement

Our work aims at exploring and enhancing the contextual reasoning capabilities of MLLMs. This work does not involve human subjects, personal data, or sensitive information. All datasets used in our experiments are publicly available benchmark datasets designed for evaluating the perceptual and reasoning capabilities of MLLMs.

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

# A. Task Definition and Construction

Our ContextReasonV-Bench comprises 16 tasks that evaluate a diverse range of contextual reasoning capabilities, including pattern induction, concept comprehension, and process tracking. Detailed descriptions and data construction process of these tasks are provided below.

## A.1. Rule-based Reasoning Tasks

**Contextual Attribute Counting (CAC)** Given a set of examples, each consisting of an image and the count of objects with a specific attribute(color, material, shape, or size), the task is to identify the target attribute and apply it to count objects with same attribute in new images. This task requires fine-grained visual recognition capabilities for precise object counting and inductive reasoning to infer the target attribute from limited examples via iterative hypothesis-validation. We utilized images from the CLEVR dataset(Johnson et al., 2017) and partition examples containing objects with the same attribute into context and query sets. For each query, four samples were randomly selected as context examples. Manual inspection was conducted on all data groups to eliminate potential confounding attributes.

**Contextual Text Transformation (CTT)** In this task, models must learn the transformation rule of the text in the red rectangle from the context and apply it to the new input. Each example consists of an image where text regions are marked by red bounding boxes, along with the transformed text. Unlike directly giving transformation rules, we focus on evaluating whether models are capable of fine-grained text recognition and rule induction. We use the images in TextVQA dataset(Singh et al., 2019) and recognize the text in the image with OpenOCR(Du et al., 2025). To isolate the impact of perceptual capability, we retain only text instances with sufficiently large bounding boxes and apply five transformation rules: capitalizing the first letter, converting all letters to uppercase, selecting unique letters, reversing words, and performing symbol substitution on specific letters. We verify that each transformed text differs from its original form to ensure the consistency of transformation rules in the context.

**Visual Logical Transformation (VLT)** This task assesses the model's ability for abstract rule induction. The abstract rules fall into two categories: visual pattern sequence transformations and numerical transformations. The former requires models to identify underlying patterns in a sequence of geometric compositions and predict subsequent elements, while the later requires models to discover mathematical relationships among spatially arranged numbers. The data samples are selected from the LogicVista(Xiao et al., 2024) and MME-Reasoning(Yuan et al., 2025) benchmarks.

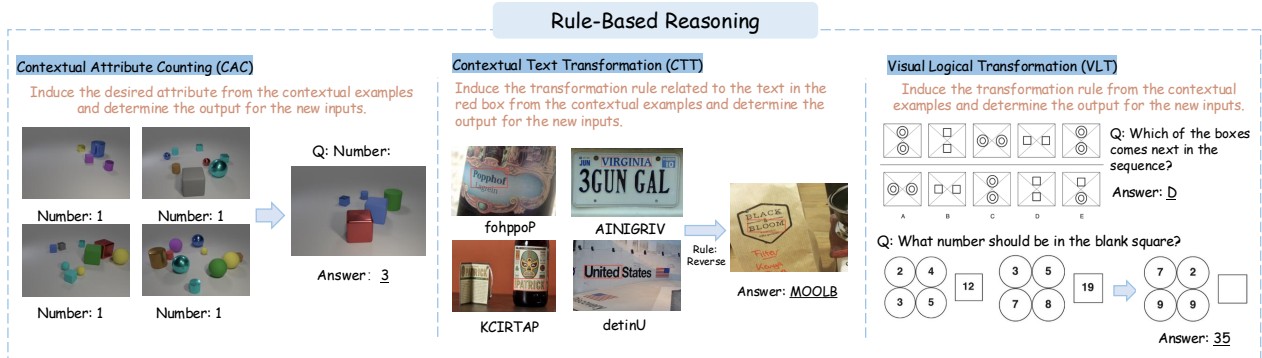

*Figure 4.* Examples of Rule-based Reasoning Tasks in ContextReasonV-Bench.

## A.2. Knowledge-based Reasoning Tasks

**Cross View Understanding (CVU)** This task requires models to comprehend sequential procedural instructions presented through multi-step demonstrations and answer queries about procedural states. Specifically, models are provided with a step-by-step tutorial consisting of multiple steps, where each step is represented by an image and corresponding textual description. The tutorial images employ an exocentric view, while the query image presents an egocentric view. This task evaluates the ability to perform cross-view action understanding based on context instructions. We sample data from multiple multi-view video datasets(Huang et al., 2024; Grauman et al., 2024; Li et al., 2024), employ CLIPScore to evaluate the semantic similarity between step annotations and different frames within the video segments of the step and select the key

frame with the highest similarity to serve as the step-wise demonstration image. To assess the model's out-of-distribution generalization capability, we train on bike repair and medical examination tasks, while evaluating on fitness and cooking scenarios in the benchmark.

**Lego Instruction Understanding (LIU)** In this task, the model is provided with step-by-step demonstrations of LEGO puzzle assembly, including the components required for each step and the state changes at each step. The model must comprehend each individual step and their sequential dependencies, and then determine the parts needed for the next step, the number of remaining steps, or the ordering of multiple steps. This task evaluates the model's ability to understand and generalize procedural instructions from a single image. We construct LEGO puzzle instructions and design question-answer pairs manually.

**Contextual Image Operation Deduction (CIOD)** This task requires the model to understand and generalize various image operations and their corresponding visual transformations of images from contextual examples, and to infer the appearance of a new image after a specific transformation is applied to it. It evaluates the capabilities of inducing both the semantic meaning and the corresponding visual changes of each operation, as well as applying the associated information to perform transformational deduction. This task consists of low-level geometric transformations (e.g., rotation, zoom, flipping) and high-level semantic modifications (e.g., background changes, style transfer, text editing). We develop a program to apply various geometric transformations to the images in COCO(Lin et al., 2014) and sample images from EMU-Edit(Sheynin et al., 2024) for high-level semantic modifications. We randomly partition the original dataset into a support set and a query set, with no overlap between the two subsets to ensure strict data independence, and sample a set of examples containing the target transformation from the support set for each query. We train the model on simpler geometric transformations and evaluate its performance on both types of operations.

**Novel Concept Recognition (NCR)** This task evaluates novel concept learning ability from limited visual examples. Models must induce category-specific visual patterns from labeled demonstrations, and then match query images to learned categories. We use fictional category names to avoid prior knowledge interference. Additionally, we deliberately misalign the ordering of images and labels within demonstrations to evaluate cross-example reasoning capbility, where models must synthesize information across multiple examples to infer correct image-label correspondences. We sample data from the Stanford Dogs (Khosla et al., 2011) and CUB dataset (Wah et al., 2011), and construct each sample in the same manner as Contextual Image Operation Deduction task mentioned above.

**Contextual Sign Language Recognition (CSLR)** This task requires models to learn letter-to-gesture mappings and identify the meaning represented by the set of gestures in the query. The sign language demonstrations in the examples and the query are performed by different individuals, thus requiring the model to possess the ability to correctly generalize the hand posture corresponding to each letter. We collect gesture images from American Sign Language involving two different individuals. To avoid interference from the model's pre-trained knowledge, we reconstruct the correspondence between gestures and letters, randomly select 20 letters to generate demonstration images, and construct query images based on synthetic words.

**Contextual Anomaly Identification (CAI)** This task requires models to detect anomalies in images and identify their specific types according to the contextual examples. It assesses the model's capacity to acquire new domain-specific knowledge from examples through inducing the visual characteristics of different anomaly types. We establish three distinct task scenarios: industrial PCB inspection, macaroni defect detection, and medical diagnosis from the VisA(Zou et al., 2022) and SMMILE(Rieff et al., 2025) datasets. The specific construction process is similar to the Contextual Image Operation Deduction task, where contextual examples with each anomaly types are randomly sampled from the support set.

**Symbolic Rotation Inference (SRI)** In this task, models are required to answer the rotation angle for query images given demonstrations each containing a pair of images before and after rotation, a symbol indicating the rotation direction (i.e., 'S' or 'N'), and a labeled rotation angle. Models must induce the rotation pattern associated with each operation symbol from examples. This task evaluates symbol-geometric transformation induction and cross-instance generalization (abstracting operation invariance across multiple samples) capabilities. We sample images from the COCO(Lin et al., 2014) dataset, construct rotational transformations at different angles via programming. For each sample, two instances are sampled from the support set for each different rotation direction.

**Manual Comprehension (MC)** This task requires models to comprehend product instruction manuals and answer related questions based on the content. Models must parse operational queries, retrieve pertinent information by understanding product features and user needs, and then generate accurate responses grounded in the documentation. We collect manuals

from the web sources and design questions based on their content manually.

**Symbolic Transformation Learning (STL)** This task requires models to learn abstract symbolic operations and perform multi-step reasoning. Each demonstration shows how a specific operation transforms a symbol sequence. Models must learn each operation's transformation and scope (all symbols or specific positions) , and then sequentially apply two operations to a query sequence to predict the final configuration. The data are selected from LogicVista(Xiao et al., 2024).

**Alphabet Decoding (AD)** This task requires the model to decode the string represented by a set of illustrations based on the specific alphabet in the context. The demonstrations include multiple alphabet tables, each establishing distinct image-to-letter correspondences. Models must identify the letters corresponding to each object in the query sequence according to the specified reference alphabet, and then concatenate them in order to form the target string. We collect alphabets from the web resources and sample three of them as the context. For each query, we construct a sequence representing the string by using the same illustrative content as found in the target alphabet.

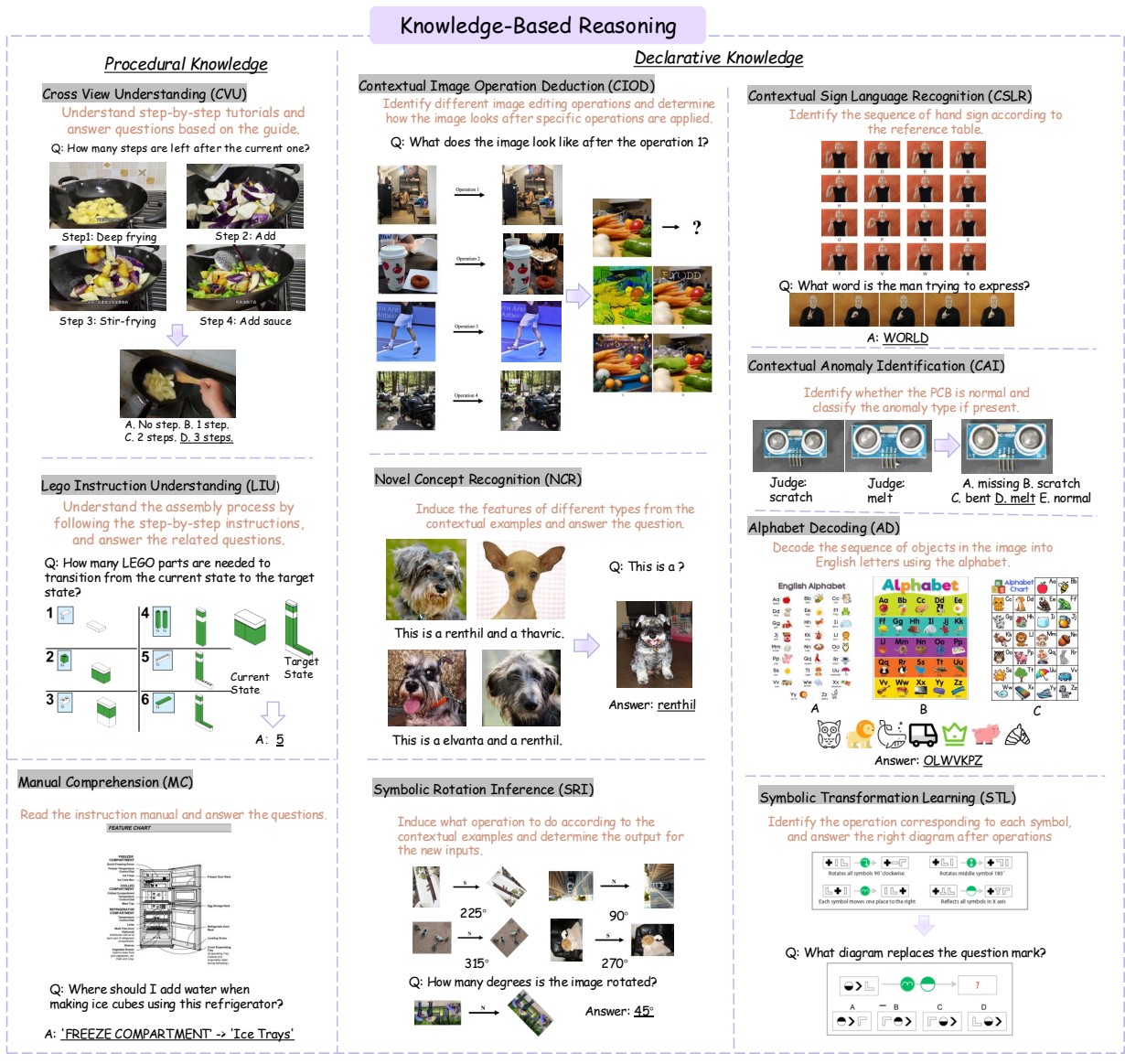

*Figure 5.* Examples of Knowledge-based Reasoning Tasks in ContextReasonV-Bench.

## A.3. Preference-based Reasoning Tasks

**Caption Preference Inference (CPI)** This task requires the model to learn user preferences from image captions paired with their corresponding ratings in the examples, and subsequently select the option that best aligns with these preferences from novel image captions. The task evaluates the model's capability to summarize fine-grained stylistic features and characteristics of image descriptions, as well as its ability to induce and generalize user preferences. In this task, we define five distinct descriptive characteristics by integrating descriptive tone (e.g., narrative tone and analytical tone) with descriptive content (e.g., accuracy and logical coherence of the content), and employ GPT-4o to generate captions satisfying each corresponding characteristic. We then establish target style preferences and instruct the model to assign ratings to each description based on its relevance to the target style. Manual inspection is conducted on the data to ensure that the generated captions conform to their respective descriptive characteristics and that the assigned ratings are reasonable.

**Preference-based Image Selection (PIS)** This task requires the model to learn visual preference patterns from the relative preferences between different image groups, and select the image that best matches the preferred style from a new candidate set. The model is provided with multiple paired image comparisons, where each pair comprises two images exhibiting distinct visual styles, along with the user's preference indication(e.g., "Preference: $A > B$" indicates preference for A). Models must aggregate information across multiple examples to infer global user preferences, and identify which candidate image best matches these learned preferences. This task evaluates the ability to abstract visual style preferences and perform transitive reasoning across multiple pairwise comparisons in the context. Images for this task are synthesized using Qwen-Image(Wu et al., 2025), conditioned on specified captions and styles.

**Preference-aware Item Recommendation (PIR)** This task requires models to learn user preferences from the historical interactions between users and items, and then recommend the items that users are most likely to choose. The historical interaction sequences includes both favorable and unfavorable interactions, from which the model needs to summarize users' preferences and identify the types of items that users dislike. The core challenge lies in accurately extracting fine-grained preferences from users' limited interaction history and identifying the alignment between candidate items and these learned preferences. We sample data from the MicroLens(Ni et al., 2025) and Video Games subset of the Amazon Reviews(Hou et al., 2024) dataset. We preprocess the user rating data from the Video Games subset by mapping ratings of 1–2 and 4–5 to binary feedback, indicating user dislike and like. For each user, all interactions are temporally ordered and divided into a history set and a candidate set. We sample a number of liked and disliked items to construct the user's historical interactions from the history set, and select one liked item and three disliked items as recommendation options from the candidate set. Finally, we manually inspect the data to ensure consistency between users' historical preferences and the characteristics of the candidate items.

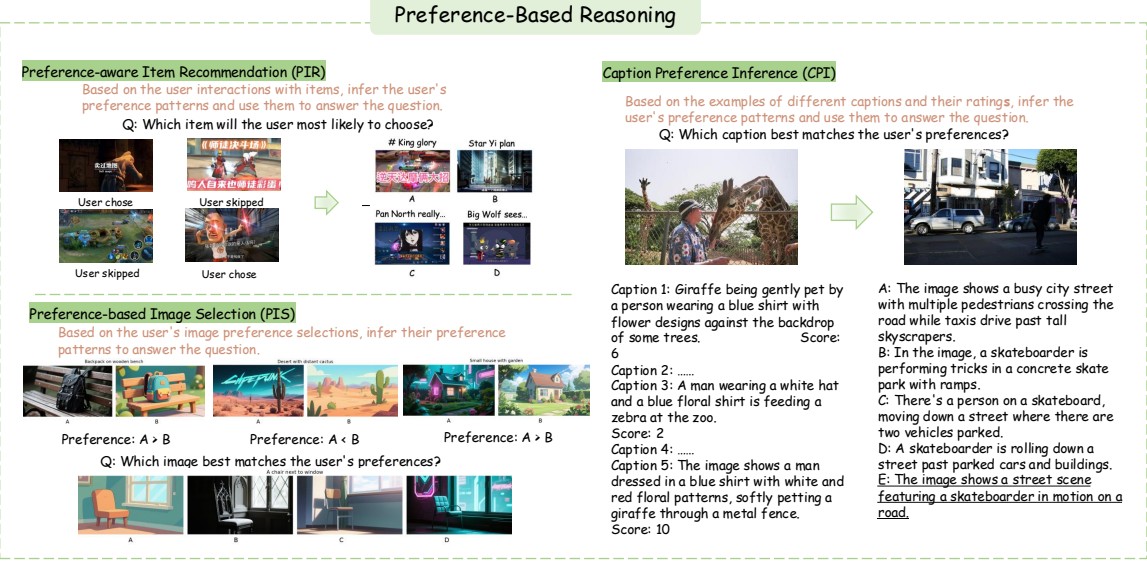

*Figure 6.* Examples of Preference-based Reasoning Tasks in ContextReasonV-Bench.

# B. Related Works

## B.1. RL for Multimodal Reasoning

Enhancing the reasoning capabilities of MLLMs enables them to integrate visual and linguistic information for more complex tasks. Extensive recent research (Chen et al., 2025a; Yang et al., 2025; Shen et al., 2025) has employed reinforcement learning to enhance the reasoning capabilities of multimodal large models, achieving promising results. Vision-R1 (Huang et al., 2025) and MM-Eureka(Meng et al., 2025) construct large-scale reasoning data and leverage cold-start initialization as well as RL training to incentivize reasoning capability. Many subsequent works(Wang et al., 2025c; Fan et al., 2025) have focused on optimizing reward design, advantage estimation and training statigies. Vision-SR1 (Li et al., 2025b) proposes a self-rewarding method to guide the intermediate reasoning process, which enhances the integration of visual perception into reasoning. R1-ShareVL (Yao et al., 2025)shares reward information across and within question variants during advantage computation for more accurate estimation of relative advantages. In contrast to these works focusing on enhancing zero-shot multimodal reasoning capabilities, our work primarily investigates how to improve the in-context reasoning capabilities of models through suitable reward design and training strategies.

## B.2. Multimodal Contextual Reasoning

As a prototypical paradigm of contextual reasoning, multimodal in-context learning has garnered widespread attention. Several benchmarks(Zong et al., 2025; Rieff et al., 2025) have been proposed to investigate the limitations of multimodal in-context learning, but they either predominantly concentrate on simple tasks amenable to straightforward pattern matching, or remain restricted to specific domains. Besides, many efforts(Awadalla et al., 2023; Chen et al., 2025b; Qin et al., 2024) have been made to enhance multimodal models' in-context learning capabilities. CAMA(Li et al., 2026) and TrueMICL (Chen et al., 2025c) adjust the attention weights toward context. $M^2IV$ (Li et al., 2025a) propose a representation engineering approach that replaces explicit demonstrations with a set of learnable vectors injected into models to improve efficiency. However, they exclusively examine in-context learning capabilities on a constrained set of general vision tasks, leaving the models' reasoning abilities in more complex contextual scenarios largely unexplored.

# C. Training Details

This section provides additional implementation details to supplement Section 4.1. The training data includes contextual reasoning tasks as well as general reasoning data sampled from Geometry3K(Lu et al., 2021) and Visulogic(Xu et al., 2025b). For tasks that overlap between the training dataset and the benchmark, we construct the tasks using different original data to ensure that no data leakage occurs during training. We perform multiple evaluations on the training data to identify and filter out samples where the model demonstrates either excessively high or low accuracy. This filtering strategy enhances training efficiency by eliminating trivial samples that contribute minimal gradients and overly difficult samples that lie beyond the model's learning frontier. The final distribution of training data in the SFT and RL stages is illustrated in Figure 7. To enhance training efficiency, we perform multiple evaluations on the training data and filter out samples where the model demonstrates either excessively high or low accuracy. We implement our RL-based methods using the VLM-R1(Shen et al., 2025) framework. The visual input resolution is limited to 401,408 pixels and the learning rate in RL phase is 1e-6. All models are trained using bf16 precision on 8 × NVIDIA A800 (80GB) GPUs.

The system prompt is as follows:

---

**System Prompt**

You are a assistant with contextual reasoning capabilities.
Your task is to extract relevant information from the context and provide a reasonable answer for each question based on the context.
Make sure to focus on the key details that relate to the question and ignore irrelevant or extraneous information.
Present the context analyzing process within the <analyze> </analyze> tags, the problem solving process within the <solve> </solve> tags, and answer within the <answer> </answer> tags, i.e., <analyze> context analyzing process here </analyze> <solve> problem solving process here </solve> <answer> answer here </answer>.

---

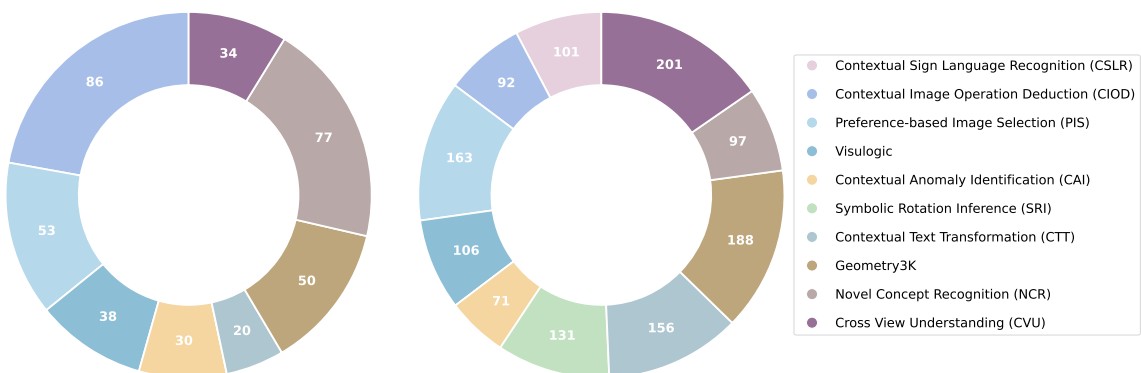

*Figure 7.* The distribution of training data in the SFT (left) and RL (right) stages.

## D. Evaluations on General Vision Task

### D.1. General Vision Benchmarks

We use the following benchmarks to evaluate the model's performance on general visual tasks.

- **MMStar** (Chen et al., 2024) evaluates models' vision-based reasoning capabilities through 1,500 carefully designed samples distributed across six core competency dimensions: coarse perception, fine-grained perception, instance reasoning, logical reasoning, science and technology, and mathematics.

- **BLINK** (Fu et al., 2024) is a comprehensive visual perception benchmark encompassing a diverse array of multi-image tasks, featuring 3,807 samples across 14 distinct task categories that span multiple levels of visual understanding, from low-level pattern matching and mid-level geometric reasoning to high-level semantic comprehension.

- **MathVerse** (Zhang et al., 2024) consists of 2.6K high-quality multi-subject math problems transformed into six versions each, designed to comprehensively assess visual diagram understanding and reasoning.

- **MMMU** (Yue et al., 2024) evaluates multimodal reasoning on multi-discipline tasks demanding college-level subject knowledge, comprising 11.5K questions spanning six core disciplines.

- **MMMU-Pro** (Yue et al., 2025) is a more robust and challenging version of MMMU benchmark, which introduces a vision-only input setting where both the question text and answer options are embedded directly within images, enabling a more rigorous assessment of MLLMs' true multimodal understanding and reasoning capabilities.

- **VisNumBench** (Weng et al., 2025) is designed to evaluate the ability of MLLMs to perceive and estimate visual numerical attributes such as angle, quantity, length, and scale. It consists of approximately 1,900 multiple-choice question-answer pairs, covering seven visual numerical attributes and four types of visual numerical estimation tasks.

- **HallusionBench** (Guan et al., 2024) is designed to diagnose two entangled failure modes in MLLMs: language hallucination, where models over-rely on language priors while ignoring visual input, and visual illusion, where visual content is misinterpreted.

- **MATH-Vision** (Wang et al., 2024) is a collection of 3,040 high-quality mathematical problems sourced from real math competitions, spanning 16 distinct mathematical disciplines and graded across 5 levels of difficulty, designed to evaluate the mathematical reasoning abilities of MLLMs.

### D.2. Analysis of General Visual Capabilities

To further analyze model performance across different levels of general visual tasks, we follow the categorization in BLINK(Fu et al., 2024) and evaluate the performance on low-level pattern matching, mid-level reasoning and high-level

visual understanding respectively. Specifically, low-level tasks (e.g. object localization) focus on low-level pattern matching, requiring models to precisely perceive pixel-level details. Mid-level tasks (e.g. relative depth estimation) require models to understand object structures, depth, and spatial relationships. High-level semantic tasks (visual similarity) focus on abstract semantic understanding, requiring models to perform style analysis and logical thinking. Our model CAR-7B achieves accuracies of 50.53%, 58.36%, and 42.37% from low-level to high-level task hierarchies, representing improvements of 1.07%, 3.27%, and 8.62% over the base model. This hierarchical improvement pattern indicates that our approach effectively enhances high-level semantic understanding and reasoning capabilities.

*Table 5.* Performance on BLINK across three hierarchical levels

| Model | Low-level | Mid-level | High-level |
|---|---|---|---|
| Qwen2.5-VL-7B | 49.46 | 55.09 | 33.75 |
| CAR-7B | 50.53 | 58.36 | 42.37 |
| Improvement | 1.07 | 3.27 | 8.62 |

## E. Case Study

We perform a systematic analysis of model reasoning trajectories to understand behavioral patterns across diverse task scenarios. By comparing the reasoning trajectories of our CAR and CAR-SFT, we find CAR demonstrates enhanced reasoning depth, logical consistency, and validation capabilities that significantly improve reasoning quality. For example, in Preference-aware Item Recommendation tasks(Figure 8), CAR conducts a more comprehensive and detailed analysis of users' historical interaction behaviors and derives fine-grained preference profiles. In contrast, the CAR-SFT merely captures superficial associations, simplistically attributing user preferences to game-related content without delving into the underlying preference patterns. Similarly, in Contextual Text Transformation tasks (Figure 9), CAR-SFT tends to misrecognize text in images and exhibit inconsistencies in the rules induced during the reasoning. Conversely, CAR enables more granular and logical analysis, and incorporates explicit validation mechanisms to ensure both correctness and consistency of the induced rules. These observations highlight the critical role of context-aware RL approach in developing robust reasoning capabilities beyond surface-level pattern recognition.

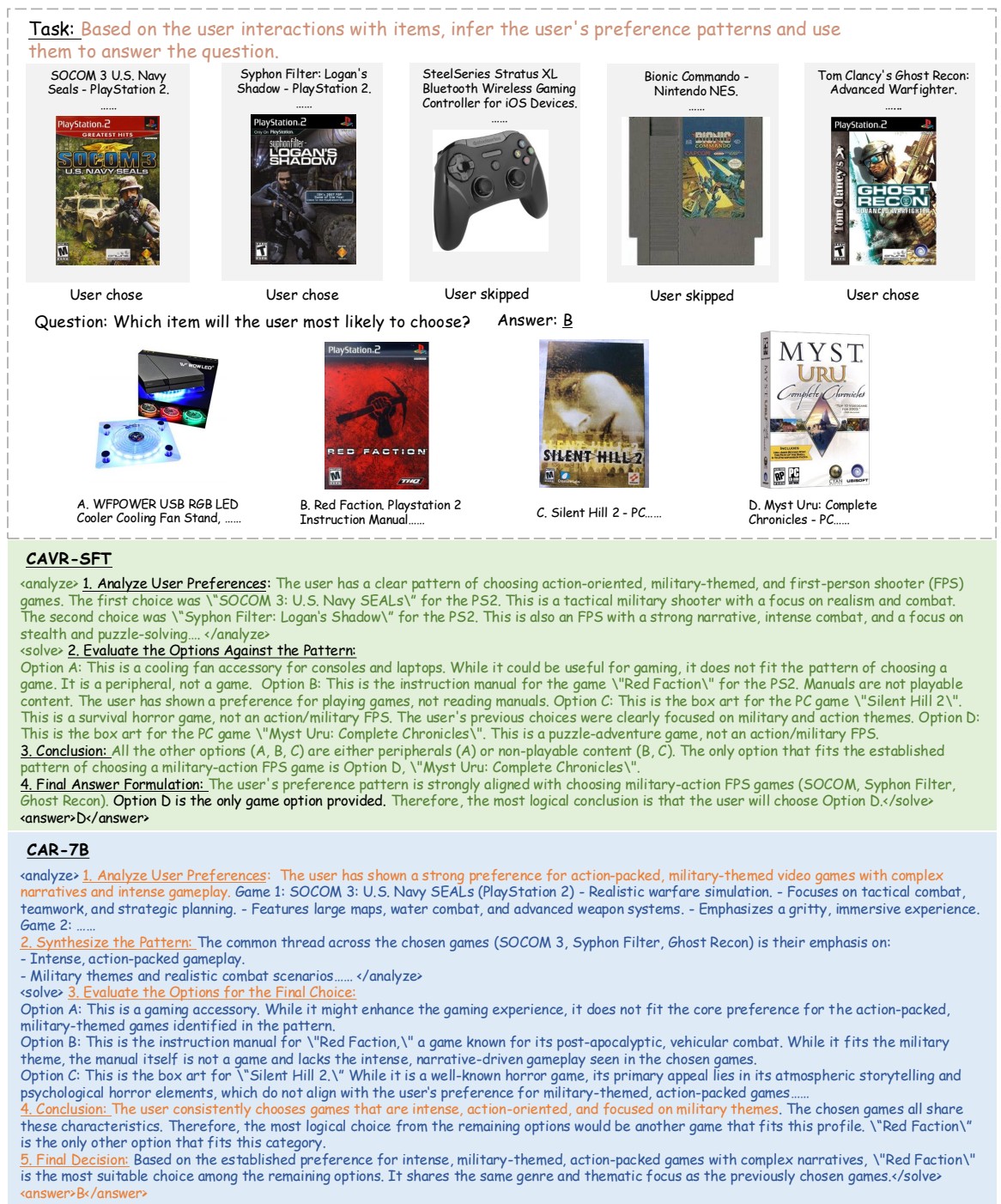

*Figure 8.* Reasoning trajectories of CAR and CAR-SFT on Preference-aware Item Recommendation task. CAR conducts a more comprehensive and detailed analysis of users' historical interaction behaviors and derives fine-grained preference profiles. In contrast, the CAR-SFT merely captures superficial associations, simplistically attributing user preferences to game-related content without delving into the underlying preference patterns.

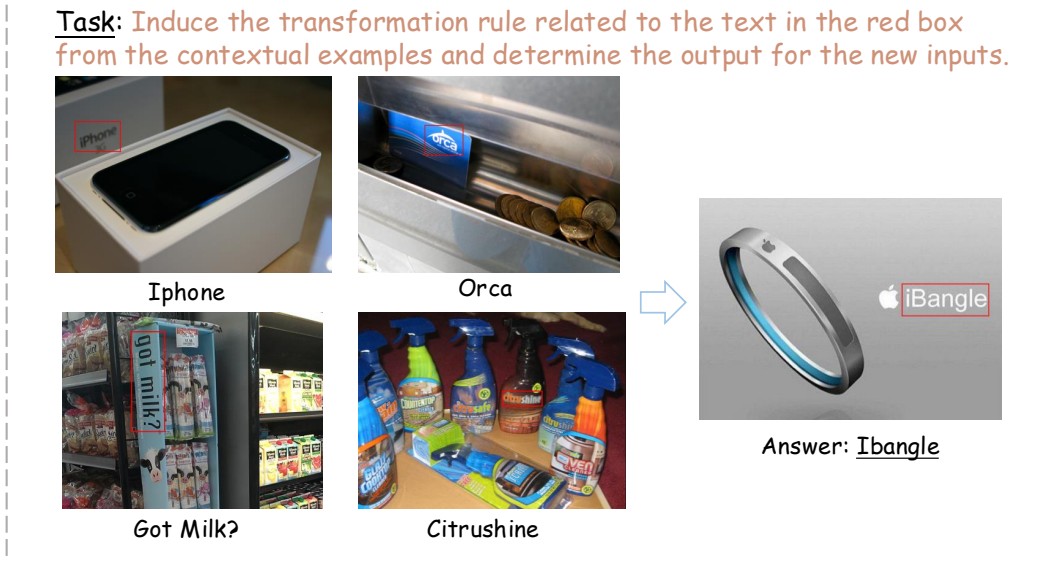

Task: Induce the transformation rule related to the text in the red box from the contextual examples and determine the output for the new inputs.

Iphone

Orca

Got Milk?

Citrushine

Answer: Ibangle

**CAR-SFT**

<analyze>1. Analyze the user's request: The user wants me to find a \"transformation rule\" from a series of image-text pairs and then apply it to a new input image.
2. Examine the provided examples: Example 1: Input Image shows \"iPhone\". The red box highlights \"iPhon\". The output is \"Iphone\". The transformation is to capitalize the first letter ('i' -> 'I') and make all other letters lowercase ('Phon' -> 'phone'). ......
Example 4: Input Image shows \"citrusafe\". The red box highlights \"citrusa\". The output is \"Citrusafe\". The transformation is to capitalize the first letter ('c' -> 'C'), the second letter ('i' -> 'I'), and the fourth letter ('a' -> 'A'), while keeping the third letter ('t' -> 't') and the fifth letter ('e' -> 'e') lowercase.
3. Induce the general transformation rule: Based on the examples, the rule appears to be: Take the text inside the red box. Capitalize the first letter of every word (including any words that are actually single characters like 's' or 'e'). Keep all other letters as they are.</analyze>
<solve> 4. Apply the rule to the new input image: Identify the text inside the red box: The text is \"iBangle\". Apply the rule: The first letter is 'i'. It should be capitalized to 'I'. The rest of the letters are 'B', 'a', 'n', 'g', 'l', 'e'. They should remain lowercase. Final Output Text: \"IBangle\". <solve>
<answer>IBangle</answer>

**CAR-7B**

<analyze> 1. Analyze the Request: The user wants me to find a transformation rule by looking at four image-text pairs (in-context examples) and then apply that rule to a new image-text pair.
2. Examine the In-Context Examples: Example 1: Input Image shows \"iPhone\" inside a red box. The output is \"Iphone\". The transformation is to make the first letter uppercase and all other letters lowercase.
 Example 2: ......
3. Induce the Transformation Rule: The common pattern across all examples is that the first letter of the entire string is converted to uppercase, while all other letters lowercase. This is a case-insensitive transformation where the first character of the input string becomes uppercase in the output.</analyze>
<solve> 4. Analyze the New Input: Input Text: \"iBangle\"
5. Apply the Induced Rule to the New Input: The input text is \"iBangle\". The first letter is 'i'. Applying the rule makes the first letter 'I'. The rest of the letters ('B', 'a', 'n', 'g', 'l', 'e') are converted into lowercase. The resulting output is \"Ibangle\".
6. Final Verification: Let's check if this rule matches the examples: \"iPhone\" -> \"Iphone\" (Correct) \"orca\" -> \"Orca\" (Correct) \"got milk?\" -> \"Got Milk?\" (Correct) \"citrushine\" -> \"Citrushine\" (Correct). The rule works consistently. The output should be \"Ibangle\".</solve>
<answer>Ibangle</answer>

*Figure 9.* Reasoning trajectories of CAR and CAR-SFT on Contextual Text Transformation task. CAR-SFT tends to misrecognize text in images and exhibit inconsistencies in the rules induced during the reasoning. Conversely, CAR enables more granular and logical analysis, and incorporates explicit validation mechanisms to ensure both correctness and consistency of the induced rules.

*Table 6.* Task Distribution of Training Data and ContextReasonV-Bench. Tasks marked with  purple  are not included in the training data, and tasks marked with  blue  include sub-tasks unseen in the training.

| Tasks | | Train | Bench | Choice | Open-ended |
|---|---|:---:|:---:|:---:|:---:|
| **Tasks** | | **Train** | **Bench** | **Choice** | **Open-ended** |
| *Rule-based Reasoning* | | | | | |
| Contextual Attribute Counting (CAC) | | | 129 | | ✓ |
| Contextual Text Transformation (CTT) | | ✓ | 100 | | ✓ |
| Visual Logical Transformation (VLT) | | | 126 | ✓ | ✓ |
| *Knowledge-based Reasoning* | | | | | |
| Cross View Understanding (CVU) | Bike Repair, Medical Examination | ✓ | – | ✓ | ✓ |
| | Fitness, Cooking | | 50 | | |
| Lego Instruction Understanding (LIU) | | | 15 | ✓ | ✓ |
| Contextual Image Operation Deduction (CIOD) | Geometric Transformations | ✓ | 150 | ✓ | |
| | Semantic Modifications | | | | |
| Novel Concept Recognition (NCR) | Stanford Dogs | ✓ | 150 | | ✓ |
| | CUB | | | | |
| Contextual Sign Language Recognition (CSLR) | | ✓ | 20 | | ✓ |
| Contextual Anomaly Identification (CAI) | $PCB_1$ | ✓ | – | ✓ | |
| | $PCB_2$, Macaroni, Medical Diagnosis | | 118 | | |
| Symbolic Rotation Inference (SRI) | | ✓ | 50 | | ✓ |
| Manual Comprehension (MC) | | | 20 | | ✓ |
| Symbolic Transformation Learning (STL) | | | 20 | ✓ | |
| Alphabet Decoding (AD) | | | 50 | | ✓ |
| *Preference-based Reasoning* | | | | | |
| Caption Preference Inference (CPI) | | | 100 | ✓ | |
| Preference-based Image Selection (PIS) | | ✓ | 100 | ✓ | |
| Preference-aware Item Recommendation (PIR) | | | 100 | ✓ | |

