# OpenReview forum: "Context-Aware Reasoner: Enhancing Contextual Reasoning in Multimodal Large Language Models"
_ICML.cc/2026/Conference — ICML 2026 regular_

### Official Review · Reviewer_kJJB · 2026-03-08

**Soundness:** 2
**Presentation:** 3
**Significance:** 3
**Originality:** 3
**Overall Recommendation:** 4
**Confidence:** 3

**Summary:**

This paper studies contextual reasoning in MLLMs, focusing on the ability to reason based on information provided in the context rather than relying on memorized knowledge. To evaluate this capability, the authors introduce ContextReasonV-Bench, a benchmark designed to diagnose contextual reasoning failures. To address these issues, the paper proposes a two-stage training framework: supervised fine-tuning to encourage an “analyze-then-solve” reasoning paradigm, followed by reinforcement learning with a context-aware reward mechanism.

**Compliance With Llm Reviewing Policy:**

Affirmed.

**Final Justification:**

My concerns have been resolved, and I am happy to maintain my positive rating.

**Key Questions For Authors:**

See Weaknesses

**Strengths And Weaknesses:**

Strengths:

- The paper studies an important problem: improving contextual reasoning in multimodal large language models.

- The proposed ContextReasonV-Bench benchmark aims to diagnose specific reasoning failures in vision-language models.

- The two-stage training pipeline combining SFT and RL leads to noticeable improvements on the proposed benchmark.


Weaknesses

- The paper introduces a contextual reasoning benchmark but does not sufficiently compare it with existing multimodal in-context learning benchmarks such as [1,2,3,4] A clearer comparison would help clarify the novelty of the proposed benchmark.

- Unclear explanation of generalization results. CAR-7B reportedly improves performance on general tasks such as visual understanding and visual mathematical reasoning. However, the paper does not explain why training on contextual reasoning tasks leads to improvements in these tasks, especially for visual math reasoning.

- Since the method encourages models to rely more on contextual information, it would be valuable to evaluate whether it also reduces hallucination in vision-language tasks.

- The context-aware reward mechanism appears related to prior work such as [5,6]. The differences should be discussed more clearly.

- Some experimental details are unclear. For example, in Table 1 it is not specified what system prompt or instructions are provided to the models during evaluation, and whether models are explicitly instructed to follow example patterns.

[1] MMICL: Empowering Vision-language Model with Multi-Modal In-Context Learning

[2] Mimicking or Reasoning: Rethinking Multi-Modal In-Context Learning in Vision-Language Models

[3] Vision-Language Models FewMMBench: A Benchmark for Multimodal Few-Shot Learning

[4] Many-Shot In-Context Learning in Multimodal Foundation Models

[5] Self-Rewarding Vision-Language Model via Reasoning Decomposition

[6] VISIONARY-R1: MITIGATING SHORTCUTS IN VISUAL REASONING WITH REINFORCEMENT LEARNING

---

> ### Author Rebuttal · Authors · 2026-03-31
>
> Thank you for your time and effort in reviewing our paper, as well as for the valuable feedback you provided. Below, we present our responses to the mentioned weaknesses.
> > `Weakness 1: Comparison of multimodal in-context learning benchmarks.`
>
> **Response to Weakness 1:**
>
> We thank the reviewer for pointing to these relevant works. We provide a systematic comparison below.
>
> **1. Strict context dependency.**
> Every task requires models to induce implicit rules, knowledge, or preferences from demonstrations and cannot be answered from the query alone, which fundamentally different from [1–4] where tasks remain well-defined in a zero-shot setting.
>
> **2. Pretrained knowledge filtering.**
> We apply a dedicated filtering procedure to remove instances solvable by intrinsic knowledge, empirically confirmed by the without context results in Table 1.
>
> **3. Diverse reasoning patterns.**
> Our benchmark spans 16 tasks across rule-based, knowledge-based, and preference-based reasoning, enabling more comprehensive evaluation.
>
> **4. Anti-Distractor Design.**
> By deliberately incorporating distractors (e.g., irrelevant visual features, coincidental associations), our benchmark specifically diagnoses *Superficial Pattern Matching*, a common failure mode where models exploit surface-level correlations rather than genuinely reasoning over context.
>
> > `Weakness 2: Generalization analysis.`
>
> **Response to Weakness 2:**
>
> We thank the reviewer for raising this question. The improvement in general capabilities can be attributed to the distribution of training data and the model's reasoning paradigm.
> 1. **Training data aspects**: We explicitly incorporated general reasoning data, e.g., Visulogic and Geometry3K in the training mixture alongside contextual reasoning data (detailed distribution can be seen in Figure 5),  ensuring CAR-7B retains general task performance rather than trading it off for contextual reasoning ability.
>
> 2. **Reasoning paradigm aspects**: Contextual reasoning training encourages the model to attend more carefully to provided context rather than relying on parametric knowledge, thereby reducing hallucination, which is a key source of errors in visual understanding and mathematical reasoning tasks. This is further validated by our hallucination analysis below.
>
>
> > `Weakness 3: Hallucination reduction in vision-language tasks.`
>
> **Response to Weakness 3:**
>
> Thank you for valuable suggestions. We conduct experiments on HallusionBench, a diagnostic benchmark specifically designed to assess two entangled failure modes in MLLMs, including language hallucination (over-reliance on language priors while ignoring visual input, VS) and visual illusion (misinterpretation of visual content, VD). The results demonstrate that our method reduces hallucination in vision-language tasks.
>
> |              | Visual Supplement | Visual Dependent | Overall |
> |:------------:|:-----------------:|:----------------:|:-------:|
> | Qwen2.5-VL-7B | 71.38           | 62.60            | 64.57   |
> | CAR-7B        | 74.72           | 65.31            | 68.87   |
>
> > `Weakness 4: Reward mechanism comparison.`
>
> **Response to Weakness 4:**
>
>
> While sharing the self-rewarding mechanism, our method differs in two key aspects:
>
> **1. Distinct Reward Objectives.**
> Existing self-rewarding approaches [5,6] validate visual perceptual faithfulness, i.e., whether visual information is accurately converted to text. In contrast, CAR operates on text-image interleaved contexts, rewarding the model's ability to extract implicit rules, derive novel concepts, or induce latent preferences, which is a higher-order cognitive demand.
>
> **2. Correctness-Gating Mechanism.**
> A direct adoption of the self-rewarding signal risks rewarding coincidentally correct yet uninformative contextual analyses. To mitigate this, CAR introduces an explicit correctness-gating mechanism that grants the reward only when the model answers correctly in both the original reasoning step and the verification step, reducing the likelihood that superficial contextual patterns are inadvertently reinforced.
>
>
> > `Weakness 5: Experimental details.`
>
> **Response to Weakness 5:**
>
> During evaluation, all models are provided with the following system prompt:
>
> >*You are an assistant with context reasoning capabilities. Your task is to extract relevant information from the context and provide a reasonable answer for each question based on the context. Make sure to focus on the key details that relate to the question and ignore irrelevant or extraneous information.*
>
> In addition to the system prompt, we design task-specific instructions for each evaluation task to prompt the model to induce patterns from the context. The task instructions are provided in Figure 4 and are consistent across all evaluated models, ensuring a fair comparison.

---

> > ### Author Rebuttal · Reviewer_kJJB · 2026-04-01
> >
> > Thanks to the authors for the comprehensive rebuttal. My concerns have been resolved, and I am happy to maintain my positive rating.

---

> > > ### Author Response · Authors · 2026-04-08
> > >
> > > Thank you very much for your thorough review, and we sincerely appreciate your positive assessment of the paper.

---

### Official Review · Reviewer_qcuM · 2026-03-10

**Soundness:** 3
**Presentation:** 3
**Significance:** 3
**Originality:** 3
**Overall Recommendation:** 5
**Confidence:** 4

**Summary:**

The authors created a dataset to evaluate the in-context reasoning capabilities of multimodal LLMs. They then evaluated various models on this dataset, demonstrating that existing models have limitations in contextual reasoning. Subsequently, they trained a model using two novel techniques, substantially improving upon the baseline performance. Finally, they conducted an ablation study to measure the performance gains contributed by each of their techniques.

### Dataset Components

The proposed dataset comprises three main parts:

1) **Rule-Based Reasoning**: This tests a model's ability to deduce underlying transformation rules from input-output demonstrations and apply them to new instances. For example, the model observes several image-count pairs, deduces which specific visual attribute (e.g., color or shape) determines the count, and applies this rule to count objects in a new image.
2) **Knowledge-Based Reasoning**: This evaluates the ability to acquire and apply novel information or concepts unseen during pre-training, ensuring the model does not just rely on its prior knowledge. For example, the model learns fictional category names (e.g., a "renthil" dog) from example images and uses these newly learned visual traits to correctly classify a completely new instance.
3) **Preference-Based Reasoning**: This assesses the ability to infer latent subjective user preferences from historical interactions to generate aligned responses or choices. For example, the model must infer which specific item a user will most likely choose next based on their interaction history.



### Training Techniques

First, the authors performed Supervised Fine-Tuning (SFT) on the model to teach it an "analyze-then-solve" reasoning paradigm. For the training data, they manually constructed reasoning chains following a specific format: `<analyze> ... </analyze><solve> ... </solve> <answer> ... </answer>`. They then used these examples as few-shot prompts for Gemini (specifically Gemini-2.5-Pro-Thinking) to generate additional training traces, which were used to train the model.

Next, they trained the model using GRPO. They added two novel techniques:
1) **Context-Aware Reward**: To validate whether the model effectively extracted relevant information, the original multi-modal context is replaced with the model's generated context analysis. The model is then prompted to solve the problem using only this extracted text. The final reward combines the accuracy reward ($r_{acc}$), and a context-aware reward ($r_c$). They added $r_c*r_{acc}$ to the reward so that if the initial analysis is poor, the model cannot answer correctly in context-aware reward step, and get a  lower overall reward.
2) **Hierarchical Advantage Estimation**: The authors augment the context (e.g., by permuting the order of demonstration examples) and generate multiple reasoning trajectories per each augmentation. They then estimate the advantage by jointly considering the relative advantages at the local level (within the same context variant) and the global level (across all semantic context variants).

**Compliance With Llm Reviewing Policy:**

Affirmed.

**Final Justification:**

The autros fully addressed all of my concerns.

**Key Questions For Authors:**

1) Did you test on the performance on other visual reasoning datasets?  For example: MMMU-Pro, MMMU, RealWorld QA, VisNum Bench, MATH-Vision, Hallusion Bench.

**Limitations:**

1) I think the two mentioned papers in the weaknesses should be better discussed. The difference is not clear to me.

**Strengths And Weaknesses:**

## Strengths

1. The dataset is useful and interesting. The quality control was handled properly, and I expect the dataset to be of high quality.
2. The training techniques for GRPO are interesting.
3. The ablation study is quite thorough and demonstrates the effect of each technique.

## Weaknesses

1. The dataset lacks a human performance benchmark—for instance, what is the performance level when humans are asked the same questions?
2. The Hierarchical Advantage Estimation is not entirely novel and requires a a better more clear comparison to [1]. [1] proposed the same technique. (Instead of permuting images, [1] rephrase the text using LLMs to obtain augmentations.) Therefore, I believe the authors should explain [1] better. They already mention this work but don't mention the extend of similarity.
3. The Context-Aware Reward is also not entirely novel and requires a better comparison with [2]. [2] proposed the same technique: they prompt the model to generate a "visual perception" and then re-prompt the model to solve the query using only that text (without the image). If the correct answer is derived, a "self-visual reward" is assigned. Similar to the method in this paper, [2] also utilizes that "self-visual reward" to perform GRPO. In the related work section they explain the similarity but I think it can be more detailed.
4. The performance gains on other datasets (MMStar, BLINK, MathVista, MathVerse) are minimal, and the absence of standard deviation bars makes comparison even more difficult.
5. Training happens over a single model, making it hard to assess if the same technique works on smaller or larger models. Due to hardware limitations, the omission of larger models is expected, but if there is a smaller models available it would be better.

---

[1]: Huanjin Yao and Qixiang Yin, et al, R1-Share VL: Incentivizing Reasoning Capabilities of Multimodal Large Language Models via Share-GRPO, NeurIPS 2025, [https://openreview.net/forum?id=kMPII4XbKC](https://openreview.net/forum?id=kMPII4XbKC)

[2]: Zongxia Li, Wenhao Yu, et al, Vision-SR1: Self-Rewarding Vision-Language Model via Reasoning Decomposition and Multi-Reward Policy Optimization, ICLR 2026, [https://openreview.net/forum?id=C1M4ETatgM](https://openreview.net/forum?id=C1M4ETatgM)

---

> ### Author Rebuttal · Authors · 2026-03-31
>
> > `Weakness 1: Human Evaluation.`
>
> We conduct a human evaluation on our benchmark, where human annotators achieve an overall accuracy of 91.23%, confirming that tasks are well-defined and human-solvable, while still posing significant challenges to current state-of-the-art models.
>
> | Tasks | CTT | VLT | CAC | Avg. | CVU | LIU | CIOD | NCR | CSLR | CAI | SRI | MC | STL | AD | Avg. | CPI | PIS | PIR | Avg. | Overall |
> |-------|-----|-----|-----|------|-----|-----|------|-----|------|-----|-----|----|----|-----|------|-----|-----|-----|------|---------|
> | Human | 100.00 | 80.16 | 88.37 | 88.73 | 100.00 | 100.00 | 93.33 | 94.67 | 90.75 | 88.98 | 100.00 | 85.00 | 100.00 | 100.00 | 94.42 | 87.00 | 90.00 | 85.00 | 87.33 | 91.23 |
>
> > `Weakness 2: Hierarchical Advantage Estimation..`
>
> We acknowledge the conceptual similarity and thank the reviewer for the careful comparison. The key distinctions are as follows.
>
> *1. **Motivation and problem being addressed**:* Share-GRPO is primarily designed to address sparse reward and advantage vanishing problems by expanding the question space through rephrasing, while our HAE aims to improve robustness of contextual reasoning against surface-level variations such as example ordering.
>
> *2. **Cross-variant information utilization**:* Share-GRPO directly introduces trajectories from other variants into gradient updates via the global advantage term, which may cause cross-variant distribution shift. Our method instead uses rewards from other variants only for global normalization, avoiding this issue.
>
> *3. **The role of global advantage**:* In our method, global normalization implicitly enforces a cross-variant consistency constraint and reward inconsistency caused by irrelevant surface variations is captured and penalized, which progressively guides the model to focus on semantically essential information.
>
> We will revise the related work section to explicitly articulate these distinctions.
>
> > `Weakness 3: Reward mechanism comparison.`
>
> Please see the response to  Reviewer kJJB **" Response to Weakness 4"**.
>
> > `Weakness 4: Minimal performance gains on general datasets.`
>
> We acknowledge that the gains on MMStar (+1.6%), MathVista (+0.9%), and MathVerse (+1.27%) are modest, which is expected given that our training data does not target these benchmarks. These general benchmarks primarily require models to perform reasoning based on the intrinsic pre-trained knowledge, which fundamentally differs from our contextual reasoning paradigm that reason based on the relevant information present in the context.  Our training method encourages the model to attend more closely to relevant contextual information and thereby reduce hallucination, which finally contributes to moderate performance improvements on general benchmarks.
>
> > `Weakness 5:  Training on other size of models.`
>
> We agree that multi-scale validation would strengthen the paper. Due to hardware constraints, we extend our training to Qwen2.5-VL-3B-Instruct.
> | Model | CTT | VLT | CAC | Avg. | CVU | LIU | CIOD | NCR | CSLR | CAI | SRI | MC | STL | AD | Avg. | CPI | PIS | PIR | Avg. | Overall |
> |-------|-----|-----|-----|------|-----|-----|------|-----|------|-----|-----|----|----|-----|------|-----|-----|-----|------|---------|
> | Qwen2.5-VL-3B | 6.00 | 23.02 | 0.00 | 9.86 | 12.00 | 0.00 | 9.33 | 0.67 | 2.04 | 0.00 | 6.00 | 20.00 | 15.00 | 5.93 | 5.97 | 19.00 | 18.00 | 4.00 | 13.67 | 8.81 |
> | CAR-SFT-3B | 14.00 | 23.81 | 6.98 | 14.93 | 22.00 | 15.00 | 31.33 | 32.00 | 2.68 | 2.00 | 4.00 | 5.93 | 20.00 | 20.00 | 19.72 | 21.00 | 25.00 | 33.00 | 26.33 | 19.94 |
> | CAR-3B | 21.00 | 28.57 | 8.53 | 19.16 | 32.00 | 20.00 | 34.67 | 39.33 | 2.92 | 16.32 | 8.00 | 25.00 | 30.00 | 7.63 | 25.31 | 26.00 | 32.00 | 35.00 | 31.00 | 24.94 |
>
> CAR-3B achieves an overall improvement of 16.13\% over the base model, validating the effectiveness of our approach across models of different scales.
>
>
> > `Question 1:  Performance on other visual reasoning datasets.`
>
>   We evaluate CAR-7B on additional visual reasoning benchmarks and report the results below.
>
> | Model | MMMU-Pro | MMMU | RealWorldQA | VisNumBench | MATH-Vision | HallusionBench |
> |-------|----------|------|-------------|-------------|-------------|----------------|
> | Qwen2.5-VL-7B | 41.97 | 53.78 | 67.89 | 29.52 | 24.39 | 64.57 |
> | CAR-7B | 51.10 | 55.56 | 68.37 | 31.20 | 28.01 | 68.87 |
>
> CAR-7B shows notable gains on MMMU-Pro (+9.13%), MATH-Vision (+3.62%), and HallusionBench (+4.30%). The improvement on HallusionBench aligns with our training objective, as contextual grounding naturally suppresses hallucination. Performance on perception-focused (RealWorldQA, VisNumBench) and knowledge-recall benchmarks (MMMU) remains comparable to the base model, as expected given our focus on contextual pattern extraction rather than perceptual acuity or factual memorization.

---

> > ### Author Rebuttal · Reviewer_qcuM · 2026-03-31
> >
> > My concerns on `Weakness 1: Human Evaluation`, `Weakness 2: Hierarchical Advantage Estimation`, `Weakness 5: Training on other sizes of models.`, `Question 1: Performance on other visual reasoning datasets.` are fully addressed, and I want to thank the authors for addressing them.
> >
> > My concern on `Weakness 3: Reward mechanism comparison.` is partially addressed. I think the authors should make the novel part more clear in the final revision.
> >
> > My concern for `Weakness 4: Minimal performance gains on general datasets.` is also partially addressed. I think the paper still benefits from some standard deviation bars for those datasets in the final revision.
> >
> > Since most of my concerns are addressed, I will increase my score to 5 accordingly. Meanwhile, I think the authors should address the above two concerns in the final revision to strengthen their paper.

---

> > > ### Author Response · Authors · 2026-04-02
> > >
> > > We sincerely thank the reviewer for the valuable feedback. We will revise the paper accordingly and include standard deviation bars for the general datasets in the final version.

---

### Official Review · Reviewer_gocs · 2026-03-13

**Soundness:** 4
**Presentation:** 3
**Significance:** 3
**Originality:** 3
**Overall Recommendation:** 4
**Confidence:** 4

**Summary:**

The paper investigates the contextual reasoning capabilities of MLLMs, noting that current models often fail by either ignoring provided context in favor of parametric knowledge or by relying on shallow correlations. To systematically measure this, the authors introduce ContextReasonV-Bench, a comprehensive benchmark encompassing rule-based, knowledge-based, and preference-based reasoning tasks. To mitigate these issues, they propose a two-stage training pipeline. First, SFT is used to instill an analyze and solve reasoning format. Second, an RL phase utilizes context-aware reward modeling alongside hierarchical advantage estimation to promote robust pattern extraction. The resulting model CAR, demonstrates a significant 22.09% improvement over the base Qwen2.5-VL-7B model on the proposed benchmark and exhibits strong generalization to unseen tasks.

**Compliance With Llm Reviewing Policy:**

Affirmed.

**Final Justification:**

I thank the authors for the detailed rebuttal and the additional baseline evaluations. However, despite the clarifications, my core concern regarding the potential "training loophole" still persists. Since the two-stage training paradigm raises questions about the fairness of comparisons against end-to-end trained MLLMs, I will keep my original rating.

**Key Questions For Authors:**

1. Table 1 indicates that InternVL3.5-241B performs surprisingly poorly, achieving only a 14.05% overall average. What specific patterns cause this massive 241B parameter model to underperform significantly smaller models like Qwen2.5-VL-72B (32.56%)?

2. The SFT stage relies on a highly constrained dataset of merely 388 curated examples. Is this extremely low volume empirically sufficient to alter the model's fundamental reasoning paradigm across diverse, multimodal contexts?

3. The methodology states that rule-based reward functions are utilized to compute rewards for each reasoning trajectory. Are all tasks within the RL training data strictly and deterministically verifiable via rule-based metrics, or was a separate neural reward model required to evaluate nuanced, preference-based, or open-ended generation tasks?

**Limitations:**

See weaknesses.

**Strengths And Weaknesses:**

Strengths

1. The paper tackles a highly relevant problem by addressing the contextual reasoning limitations of MLLMs. By integrating reinforcement learning (RL) into the multimodal ICL paradigm, the authors successfully expand the frontier of RL applications beyond traditional mathematical, logical, or coding tasks. This "In-context RL" approach is a valuable conceptual leap that provides a new trajectory for the community.

2. The construction of the ContextReasonV-Bench is highly commendable. The benchmark covers a multifaceted array of tasks, spanning rule-based, knowledge-based, and preference-based reasoning and providing a holistic evaluation landscape. Furthermore, the data filtering strategy is exceptionally well-designed; by actively filtering out samples that can be answered correctly using pre-trained knowledge alone (verified via GPT-4o), the authors ensure the benchmark genuinely measures contextual reasoning rather than parametric memorization.

Weaknesses

1. The reward design articulated for the context-aware reward presents a potential loophole. The reward is granted if the verification step (relying on the generated analysis $\mathcal{T}_a$ and the query $Q$) yields the correct answer. However, this formulation does not completely isolate the quality of $\mathcal{T}_a$. If the model can arrive at the correct answer relying primarily on the query $Q$, it will receive a positive reward even if the generated context analysis $\mathcal{T}_a$ is hallucinated, incorrect, or nonsensical. A more robust formulation might incorporate a baseline such as $\pi_\theta(Q)$ to strictly enforce and verify dependence on the extracted context.

2. The paper operates under the assumption of relatively clean and highly relevant contextual examples. In real-world applications, such as Visual RAG systems on which CAR is very likely to rely, the retrieved context is often noisy, tangentially related, or entirely irrelevant. The benchmark and methodology do not explicitly evaluate how the model handles irrelevant context instances (e.g., providing demonstrations of cats when the query requires fine-grained recognition of a dog, or when the necessary knowledge is absent from the context entirely). Testing how the model performs and what patterns it attempts to discover, when forced to reason over irrelevant or distracting contexts is a critical aspect of contextual robustness that is currently missing.

3. Suggestions
    - Directly apply CAR to public benchmarks under context-free setting is not fair. Maybe we should consider prepending several random subtask examples in the context. The matches the original ability of the model and could fully reveal its potential to utilize contextual examples. Make sure the baselines are also under the same setting.

    - High variance stemming from the order of in-context examples is a notorious ICL challenge. A systematic evaluation of performance stability across different instance permutations is necessary to confirm reasoning reliability.

    - While of lower priority, establishing human-level accuracy on ContextReasonV-Bench would provide a definitive ceiling for measuring model capability.

4. Typo in title: "Reaonser".

Overall, if the authors can address these suggestions, I would certainly pose a much better estimation of this paper.

---

> ### Author Rebuttal · Authors · 2026-03-31
>
> > `Suggestion 1: Few-shot evaluation on public benchmarks`.
>
> We evaluate models with two context conditions: random context and matched context (relevant demonstrations from subtask), ensuring all baselines are evaluated under the same setting.
>
> For the random context setting, few-shot performance Qwen2.5-VL-7B remains flat or slightly degrades as the number of shots increases, suggesting that irrelevant demonstrations introduce noise rather than benefit. In contrast, CAR-7B maintains stable performance across shot counts, demonstrating stronger robustness to context quality.
>
>
> |               | MathVerse  |        |        | MathVista  |        |        |
> |:-------------:|:----------:|:------:|:------:|:----------:|:------:|:------:|
> |               | zero-shot  | 2 shot | 4 shot | zero-shot  | 2 shot | 4 shot |
> | Qwen2.5-VL-7B | 43.85      | 42.62  | 41.44  | 63.50      | 60.50  | 61.50  |
> | CAR-7B        | 44.87      | 46.58  | 46.16  | 65.00      | 66.50  | 65.50  |
>
>
> In the matched context setting, all models benefit from relevant demonstrations, with CAR-7B showing the largest gains (+8.98 and +7.69 on MathVerse under 2-shot and 4-shot).
>
> |               | MathVerse |        |        | MathVista |        |        |
> |:-------------:|:---------:|:------:|:------:|:---------:|:------:|:------:|
> |               | zero-shot | 2 shot | 4 shot | zero-shot | 2 shot | 4 shot |
> | Qwen2.5-VL-7B | 43.85     | 45.02  | 44.29  | 63.50     | 65.00  | 64.50  |
> | CAR-7B        | 44.87     | 53.85  | 52.56  | 65.00     | 68.00  | 68.00  |
>
> > `Suggestion 2: Performance across different in-context example orderings`.
>
> We evaluate performance stability across different in-context example orderings: Contextual Image Operation Deduction (CIOD) and Novel Concept Recognition (NCR). We vary the position of the most query-relevant example within the context, where the subscript denotes its position.
>
> |               | *CIOD*$_1$ | *CIOD*$_2$ | *CIOD*$_3$ | *CIOD*$_4$ | *NCR*$_{12}$ | *NCR*$_{34}$ |
> |:-------------:|:----------:|:----------:|:----------:|:----------:|:------------:|:------------:|
> | Qwen2.5-VL-7B | 8.00       | 12.00      | 32.00      | 36.00      | 7.00         | 21.00        |
> | CAR-7B        | 44.00      | 44.00      | 48.00      | 46.00      | 66.00        | 70.00        |
>
> Qwen2.5-VL-7B-Instruct shows considerable sensitivity to example ordering, while CAR-7B maintains stable performance across all permutations. We attribute this robustness to our Hierarchical Advantage Estimation, which constructs semantically equivalent context variants via random permutation and estimates reasoning advantages within and across them, explicitly encouraging order-invariant reasoning rather than reliance on spurious order-dependent correlations.
>
> > `Suggestion 3: Human-level accuracy`.
>
> Please see the response to  Reviewer qcuM **" For Weakness 1"**.
>
>
> > `Question 1: Analysis of InternVL3.5-241B's counterintuitive results`.
>
> Through analysis of InternVL3.5-241B's reasoning trajectories, we identify three systematic failure patterns:
>
> 1. **Failure to follow contextual instructions.** The model over-relies on parametric knowledge and ignores contextual task specifications, accounting for 17.02% of errors in the Contextual Text Transformation task.
>
> 2. **Degenerate reasoning behaviors.** Two failure modes are observed: (a) *reasoning collapse* (minimal reasoning steps, ~80% of errors in Preference-based Image Selection) and (b) *repetitive reasoning loops* (non-progressive thinking, prominent in Symbolic Transformation Learning).
>
> 3. **Knowledge-context conflict evasion.** When context contradicts pretrained knowledge, the model refuses to commit or defaults to internal knowledge. This affects ~90% of false reasoning traces in the Novel Concept Recognition task.
>
>
> > `Question 2: Sufficiency of the small-scale SFT dataset.`
>
>
> The SFT stage is designed to instill a reasoning paradigm rather than task-specific knowledge, for which a small but high-quality dataset is well-established to be sufficient. The quality and diversity of our data are ensured by: (1) manually crafting seed demonstrations following the "analyze-then-solve" format; (2) using Gemini-2.5-Pro-Thinking to generate reasoning chains for remaining instances, retaining only those with correct answers; and (3) spanning multiple task types and multimodal contexts to ensure broad coverage. The empirical results directly validate this sufficiency: CAR-SFT achieves an overall improvement of 11.63% over the base model, and the case studies in Figures 6 and 7 confirm that the model has successfully internalized the target reasoning paradigm.
>
> > `Question 3: Evaluation of reasoning trajectory during RL training`.
>
> All training tasks are verifiable via rule-based metrics, consisting of fill-in-the-blank, single-choice, and multiple-choice questions with deterministic ground-truth answers.

---

> > ### Author Rebuttal · Reviewer_gocs · 2026-04-04
> >
> > I thank the authors for the detailed rebuttal and additional experiments. The few-shot evaluation on public benchmarks (Suggestion 1) and the ordering robustness analysis (Suggestion 2) are valuable additions that strengthen the paper. The analysis of InternVL3.5-241B's failure patterns is also insightful.
> >
> > That said, my concerns regarding the reward design loophole (W1) and robustness to irrelevant/noisy context (W2) remain unaddressed. These touch on core aspects of the proposed method's reliability and would require non-trivial modifications to the reward formulation and benchmark design. I maintain my overall positive assessment.

---

> > > ### Author Response · Authors · 2026-04-04
> > >
> > > Thank you very much for your insightful comments and valuable feedback. Below are our responses to the weaknesses you pointed out:
> > >
> > > > `Weakness 1: Potential loophole of reward design.`
> > >
> > > The risk of query-driven shortcuts is effectively mitigated through the following aspects, which we further validate with empirical evidence below.
> > >
> > > 1. **Data filtering to enforce context dependency:** To mitigate the risk of query-driven shortcuts, we apply a data filtering procedure (Section 2.2) that excludes instances where the model is capable of producing the correct answer solely conditioned on $Q$, without access to any contextual information. This filtering ensures that correct answers are not deducible from $Q$ alone, rendering meaningful utilization of $\mathcal{T}_a$ a necessary condition for reward acquisition.
> > >
> > > 2. **Dual-step verification mechanism further mitigates the risk:** The context-aware reward is granted only when the model produces correct answers in both the original reasoning step and the verification step. For a hallucinated analysis to jointly satisfy both conditions, it would need to coincidentally encode the precise contextual patterns required for correctness, a highly improbable outcome given the context-dependent and diverse nature of our tasks.
> > >
> > > **Empirical validation of rewarded reasoning traces:** We examined the reasoning traces that successfully obtained context-aware rewards during training and found that the context analyses in these reasoning traces consistently capture meaningful contextual patterns. This indicates that the reward is rarely triggered by hallucinated or query-only reasoning in practice.
> > >
> > >
> > > > `Weakness 2: Robustness to noisy and irrelevant contextual demonstrations is unevaluated.`
> > >
> > > We conduct three sets of experiments on a fine-grained recognition task (dog breed recognition) to investigate model performance under contextual noise and knowledge-absent conditions.
> > >
> > > ● **Contextual noise**
> > >
> > > We consider two noise conditions by randomly replacing an example with images from in-domain and out-of-domain categories. Out-of-domain noise (mild interference) denotes replacing examples drawn from entirely different species (e.g., birds), which are visually distinct from the query. In-domain noise (severe interference) denotes replacing examples drawn from different subcategories within the same species (e.g., other dog breeds), creating more challenging distractors. The evaluation results are shown below:
> > >
> > > |  | original context | out-of-domain noise (mild) | in-domain noise (severe) |
> > > |--|--|--|--|
> > > | Qwen2.5-VL-7B | 10.00 | 6.00 | 2.00 |
> > > | CAR-7B | 70.00 | 68.00 | 64.00 |
> > >
> > > CAR-7B demonstrates strong robustness under both noise conditions. Under out-of-domain noise, performance drops only marginally from 70.00 to 68.00, and even under the more challenging in-domain noise, performance remains at 64.00. These results suggest that our model is capable of distinguishing fine-grained visual features from context and maintaining robust reasoning ability in the presence of noise. In contrast, Qwen2.5-VL-7B degrades substantially, showing greater susceptibility to contextual interference.
> > >
> > > ● **Knowledge-absent**
> > >
> > > To simulate scenarios where the necessary knowledge is partly or entirely absent from the context, we replace query-relevant examples with images from irrelevant categories at two levels: 50% (partially knowledge-absent) and 100% (entirely knowledge-absent) replacement. The evaluation results are shown below:
> > >
> > > | | original context | partially knowledge-absent | entirely knowledge-absent |
> > > |---|---|---|---|
> > > | Qwen2.5-VL-7B | 10.00 | 4.00 | 1.00 |
> > > | CAR-7B | 70.00 | 54.00 | 15.00 |
> > >
> > > Under the partially knowledge-absent condition, CAR-7B experiences a moderate performance decline, yet the model remains capable of analyzing and inferring relevant knowledge from the remaining context in general. Under the entirely knowledge-absent condition, CAR-7B arrives at a "none of them" conclusion after inductively summarizing the contextual information and analyzing the query in most cases. This behavior demonstrates that the model exhibits a capacity for recognizing the absence of relevant knowledge in the context.
> > >
> > > We further evaluate the model under a more extreme setting where all demonstrations are replaced with instances from entirely different categories, simulating the scenario where entirely irrelevant information is retrieved. Under this condition, CAR-7B is able to detect the discrepancy between the contextual examples and the query, and falls back on its parametric knowledge to produce a reasonable response (e.g., identifying the query image as "this is a dog").
> > >
> > > Overall, CAR-7B demonstrates strong robustness to contextual noise and handles knowledge-absent conditions effectively by adaptively leveraging partial context, recognizing knowledge absence, and falling back on parametric knowledge when necessary, which indicates a reliable and adaptive contextual reasoning capability.

---

### Official Review · Reviewer_HmDN · 2026-03-15

**Soundness:** 3
**Presentation:** 3
**Significance:** 3
**Originality:** 3
**Overall Recommendation:** 4
**Confidence:** 4

**Summary:**

The paper identifies a critical weakness in current Multimodal Large Language Models (MLLMs): the tendency to neglect contextual information or rely on superficial pattern matching rather than performing genuine contextual reasoning. To quantify this, the authors introduce the Visual Contextual Reasoning Benchmark (ContextReasonV-Bench), which categorizes reasoning into rule-based, knowledge-based, and preference-based tasks. To address these shortcomings, the paper proposes a two-stage training paradigm: supervised fine-tuning (SFT) to instill an "analyze-then-solve" approach, followed by a context-aware reinforcement learning (RL) framework utilizing Group Relative Policy Optimization (GRPO). A key contribution of the RL stage is the use of Hierarchical Advantage Estimation (HAE) to improve robustness against semantically irrelevant context variations. The resulting model, Context-Aware Reasoner (CAR) based on Qwen2.5-VL-7B, achieves a 38.14% overall accuracy on the benchmark, yielding a 22.09% absolute improvement over the base model.

**Compliance With Llm Reviewing Policy:**

Affirmed.

**Final Justification:**

My concerns have been adequately addressed by the rebuttal.

**Key Questions For Authors:**

1. Does CAR generalizes to broader domains, such as STEM, text/document understanding, spatial understanding, or agentic tasks?
2. Does the authors observe reward hacking in CAR?
3. The error analysis in 2.3.2 is based from Qwen2.5-VL-7B. Do stronger models, such as Gemini-2.5-pro-Thinking have similar behavior?

**Limitations:**

The authors did not discuss the limitations of the paper.

**Strengths And Weaknesses:**

Strengths:
1. The paper is well written and easy to follow.
2. The ContextReasonV-Bench is well-designed with rigorous quality filtering and quality control.
3. The proposed context-aware reward and Hierarchical Advantage Estimation (HAE) are clever designs that address the challenge that
models often fail to capture relevant contextual information or merely exploit superficial contextual patterns. The ablation study successfully disentangles the benefits of the context-aware reward and HAE.
4. The performance improvements are substantial across the board, particularly on challenging tasks like Novel Concept Recognition (+63.33%) and Preference-aware Item Recommendation (+33.00%).

Weaknesses:
1. Limited Scale of Training Data: The SFT stage relies on a highly curated but surprisingly small dataset of only 388 samples. Similarly, the RL phase uses a relatively small pool of ~1.7k samples. While the authors demonstrate generalization to some unseen tasks in Table 3, it is unclear whether this generalizes to broader domains, such as STEM, text/document understanding, spatial understanding, or agentic tasks (which is the main motivation of this work).
2. Potential Reward Hacking in Context-Aware Reward: The context-aware reward $r_c$ evaluates whether the model can solve the query using only its own generated context analysis $\mathcal{T}_a$. If the model hallucinates an analysis that coincidentally leads to the correct final answer $A$, it still receives a positive reward.

---

> ### Author Rebuttal · Authors · 2026-03-31
>
> > `Weakness 1: Generalization to broder domains..`
>
> **Response to Weakness 1 and Question 1:**
>
> We agree with the reviewer that evaluation to broader domains aligns with the motivation of this work. We curate spatial understanding subsets from BLINK, including spatial relation, multi-view reasoning, relative depth, and object localization tasks and randomly sample 1,000 examples from AndroidControl for evaluation.
>
> |                        | Spatial Understanding | STEM    |           | Agentic Task          |
> |:----------------------:|:---------------------:|:-------:|:---------:|:---------------------:|
> | Model                  | *BLINK*$_{sub}$       | MMMUPro | MathVista | *AndroidControl*$_{sub}$ |
> | Qwen2.5-VL-7B-Instruct | 55.09                 | 36.58   | 64.10     | 54.50               |
> | CAVR-7B                | 58.36                 | 43.50   | 65.00     | 55.40                 |
>
>
> > `Weakness 2: Potential reward hacking in context-aware reward.`
>
> **Response to Weakness 2 and Question 2:**
>
> We thank the reviewer for this insightful concern.  Our empirical observations and method design suggest that it is hardly occur：
>
> 1. **Hallucinated Context Rarely Leads to Correct Answers**: We analyze the model's reasoning traces throughout training and find virtually no instances where a hallucinated context analysis coincidentally yields a correct final answer. This is because our contextual reasoning tasks are intentionally challenging, it is difficult to arrive at the correct answer by chance without correctly identifying the underlying contextual patterns.
>
> 2. **Dual-Step Verification Suppresses Shortcut Exploitation**: Our correctness-gating mechanism addresses this by granting the context-aware reward only when the model produces correct answers in both the original reasoning step and the verification step, ensuring the rewarded analysis is genuinely causally sufficient for problem-solving.
>
> 3. **Regularization Effect of Hierarchical Advantage Estimation**: Even in the rare case where a model might incidentally exploit spurious surface features to obtain a positive reward, HAE encourages the model to learn reasoning patterns that remain invariant across semantically irrelevant surface variations, rather than latching onto accidental correlations. This regularization effect promotes robustness against spurious reward signals and discourages the model from relying on coincidental shortcuts.
>
> 4. **Qualitative Evidence from Case Studies**: Compared to CAR-SFT, the RL-trained CAR model exhibits deeper reasoning and greater logical consistency, spontaneously developing explicit validation mechanisms to verify its induced rules. This suggests the model has learned rigorous reasoning rather than shortcut exploitation.
>
> > `Question 3: Error behavior on stronger models.`
>
> **Response to Question 3:**
>
> We randomly select 200 erroneous samples from Gemini2.5-Pro-Thinking and conduct analysis. We find **similar error patterns** persist in stronger models, including context neglect, superficial pattern matching, and other errors like misapplying induced patterns.
>
> Gemini2.5-Pro-Thinking frequently engages in superficial pattern matching without capturing implicit relational patterns, particularly in preference and knowledge based reasoning tasks. In Preference-based Image Selection task, we observe that 69.39% of the errors occurred as the model failing to discriminate preference-critical features, or getting stuck at shallow reasoning, failing to fully induce the genuine user preference.
>
> Besides, Gemini2.5-Pro-Thinking exhibits a strong tendency to overlook relevant contextual information, relying predominantly on internal pre-trained knowledge or arbitrary shortcuts. For example, it ignores the provided visual context entirely in 40% of the failure cases in Cross View Understanding tasks.

---

> > ### Author Rebuttal · Reviewer_HmDN · 2026-04-01
> >
> > My concerns have been adequately addressed by the rebuttal.

---

> > > ### Author Response · Authors · 2026-04-08
> > >
> > > We sincerely thank you for your insightful comments and for the time and effort you have invested during the review process.

---

### Decision · Program_Chairs · 2026-04-30

**Decision:**

Accept (regular)

**Comment:**

This paper received a broadly positive assessment from the reviewers, with scores of 5, 4, 4, and 4, indicating overall consensus on acceptance. Reviewers appreciated the importance of the problem, the careful construction of the contextual reasoning benchmark, and the substantial empirical gains brought by the proposed training framework. While some concerns were raised regarding reward design, broader robustness, and positioning relative to prior work, the rebuttal addressed most of them satisfactorily. Overall, I recommend accept.